# Circulating causal protein networks linked to future risk of myocardial infarction

Sean Bankier [1,2,10], Valborg Gudmundsdottir [2,3,10], Thorarinn Jonmundsson [3], Heida Bjarnadottir [3], Joseph Loureiro [4], Lingfei Wang [5], Elísabet A. Frick [2], Nancy Finkel[4], Anthony P. Orth[6], Thor Aspelund [2], Lenore J. Launer [7], Johan L. M. Björkegren[8], Lori L. Jennings [4], John R. Lamb[9], Vilmundur Gudnason [2,3,11], Tom Michael [1,11] ✉ & Valur Emilsson [2,3,11] ✉

Variations in blood protein levels have been linked to numerous complex diseases, including cardiovascular conditions. These associations highlight the intricate interplay between local and systemic factors in cardiovascular disease development, emphasizing the need for a comprehensive, systems-level understanding of its etiology. To address this, we develop a causal network inference framework using data from one of the largest serum proteomics studies to date, comprising measurements of 7523 serum proteins in the prospective, population-based Age, Gene/Environment Susceptibility-Reykjavik Study (AGES) cohort of 5376 older adults. Using *cis*-acting protein quantitative trait loci (pQTLs) as instrumental variables within a causal inference framework designed to mitigate hidden confounding, we identify 185 high-confidence causal serum protein subnetworks collectively interacting with 5611 targets. Several subnetworks, many forming hierarchical frameworks of directional relationships, are significantly associated with multiple cardiometabolic traits and with future risk of myocardial infarction and its long-term complication, heart failure.

Atherosclerotic cardiovascular disease (ACVD) is the leading cause of age-standardized deaths globally[1]. While lipid-lowering treatments have been shown to reduce the risk of ACVD[2], residual risk persists[3–5], underscoring a significant and unmet medical need. Early coronary atherosclerosis, which advances to coronary artery disease (CAD), is the primary cause of ACVD. In its most advanced stage, coronary artery plaques may rupture, manifesting clinically either through myocardial infarction (MI) or stroke. The rate of coronary plaque growth is influenced by various systemic factors across multiple organ systems, such as the immune system driving systemic inflammation, the liver

regulating lipoprotein metabolism, and adipose tissue and skeletal muscle contributing to the development of obesity and type 2 diabetes (T2D)[6]. Other contributing factors are endocrine signaling and hemodynamic processes[6]. The rate at which CAD progresses depends on the interplay between these systemic factors, ultimately leading to plaque rupture, thrombus formation, and MI[6,7]. The complex etiology of ACVD is shaped by both genetic and lifestyle factors[8], which are mediated through interactions between multiple organ systems[9]. The molecular disruptions across organs that contribute to ACVD have primarily been studied in isolated pathways using model systems. While these studies

[1]Computational Biology Unit, Department of Informatics, University of Bergen, Bergen, Norway. [2]Icelandic Heart Association, Holtasmari 1, Kopavogur, Iceland. [3]Faculty of Medicine, University of Iceland, Reykjavik, Iceland. [4]Novartis Biomedical Research, Cambridge, MA, USA. [5]University of Massachusetts Chan Medical School, Worcester, MA, USA. [6]Novartis Biomedical Research, 10675 John Jay Hopkins Drive, San Diego, CA, USA. [7]Laboratory of Epidemiology and Population Sciences, National Institute on Aging, Bethesda, MD, USA. [8]Department of Medicine, Karolinska Institutet, Karolinska Universitetssjukhuset, Huddinge, Sweden. [9]Monoceros Biosystems, 12636 High Bluff Drive, Suite 400, San Diego, CA, USA. [10]These authors contributed equally: Sean Bankier, Valborg Gudmundsdottir. [11]These authors jointly supervised this work: Vilmundur Gudnason, Tom Michoel, Valur Emilsson. ✉e-mail: tom.michoel@uib.no; valur@hjarta.is

offer valuable insights into disease etiology and treatment, they fail to capture the systemic complexity of ACVD. In other words, the rate at which ACVD progresses to become clinically significant depends on the interplay of both local (cardiovascular) and systemic (non-cardio-vascular) factors, an aspect of the disease that is often overlooked. Thus, a broader systems-level approach is required to obtain a more holistic understanding of its etiology.

Tissue-specific and cross-tissue transcriptional networks have been established as both undirected networks[10–14] and directed networks[15,16], operating within and across various tissues, and demonstrating robust associations with complex diseases. In contrast to undirected co-regulatory networks, directed networks have the potential to differentiate between causal and reactive nodes and elucidate how these causal nodes propagate their effects[17]. Gene expression quantitative trait loci (eQTLs) have been employed as genetic instruments in causal inference analysis and as priors in reconstructing transcriptional Bayesian networks[15]. These variants offer an effective means of natural perturbation to infer causal relationships between genes and higher-order phenotypes like disease, and even between genes themselves[12,14]. This has been well documented by the explosion of Mendelian Randomization (MR) studies that use genetic variants as instrumental variables for both molecular and higher-order phenotypes[18]. Although reconstructing directed causal networks has traditionally been computationally intensive and limited to small-scale systems, recent advancements have significantly enhanced performance[16,19–21]. These improvements have made the process orders of magnitude faster, enabled the coverage of a larger proportion of variance in the data, and proven especially effective when both genotype and molecular node data are available for the same sample[19,20].

Proteomics has recently advanced to the point where high-throughput measurements allow for the analysis of thousands of proteins from a single tissue or biofluid sample in large population studies[22,23], exposing the depth and complexity of the plasma and serum proteomes. These recent advancements have largely been driven by the highly sensitive aptamer-based affinity methods. In fact, comparisons between various proteomics platforms highlight the superior performance of the aptamer-based platform, especially in terms of detection precision and sensitivity[24]. Using this approach led to the identification of the first human protein co-regulatory network, reconstructed from the analysis of 4137 serum proteins measured in 5457 older adults of the prospective, population-based AGES cohort[23]. Furthermore, we demonstrated that the network modules for circulating proteins are under strong genetic control and are linked to a broad range of past, current and future disease states[23]. Notably, the structure and composition of these serum protein networks are stable and have been validated in other body fluids, such as in plasma[25] and cerebrospinal fluid[26]. Finally, unlike solid tissue networks, the serum protein network consists of modules incorporating proteins synthesized by many, if not all, tissues across the body[23,27].

In this study, we describe the reconstruction of a directed circulating Causal Protein Network (CPN) within the AGES cohort using an expanded dataset comprising 7523 serum proteins encoded by 6586 genes. Here, cis-acting protein quantitative trait loci (pQTLs) were employed as instrumental variables to highlight causal interactions between protein pairs. Applying various filters, including network size and genome linkage disequilibrium (LD) between instrumental variables, we identified 185 CPN subnetworks encompassing at least 10 protein members and interacting with a total of 5611 target serum proteins. The CPN subnetworks were examined for their relationships with each other and tested for associations with various ACVD-related outcomes. We highlight the subnetworks with the strongest associations to incident MI, along with cardiometabolic traits that contribute to the risk of ACVD.

## Results

### Study population and analysis overview

This study builds on the population-based prospective AGES cohort of older adults ($N = 5764$, mean age $76.6 \pm 5.6$ years, age range 66–98 years, 57% female). The cohort is richly annotated with data on disease risk factors, endpoints, comorbidities, genotype information, and deep serum proteomics[23,28]. Table 1 displays the demographic, biochemical, clinical, physiological, and anthropometric data, as well as cardiovascular imaging measurements of participants in the AGES study, analyzed for 7523 serum proteins and stratified by incident MI after excluding all prevalent MI cases. The follow-up period for newly diagnosed MI patients extended up to 12 years from baseline, where person-years of follow-up were calculated from the first AGES visit until the date of diagnosis, death, or the end of the follow-up period, with a median of 5.6 [2.8, 8.2] years for the incident MI group (Table 1). As anticipated, several measures associated with an increased risk of an MI event were significantly altered in the incident MI group compared to the non-MI group (Table 1). These include a higher prevalence of metabolic syndrome (MetS) and type 2 diabetes (T2D), as well as elevated coronary artery calcium (CAC) and carotid artery plaque burden (Table 1). Furthermore, 25.7% of the incident MI group developed heart failure (HF), compared to 4.6% of the non-MI group ($P = 3 \times 10^{-55}$) (Table 1). Figure 1 presents an overview of the study and its workflow, including the reconstruction of the circulating CPN within the AGES study, as well as key disease endpoints and cardiometabolic traits associated with ACVD that are examined in the present study. Additional details on the data and analyses are provided in Supplementary Fig. S1.

### Reconstruction of the circulating causal protein network

We reconstructed a global network of circulating proteins using a causal inference framework, inferring edges from pairwise protein relationships, with cis-acting pQTLs at a false discovery rate (FDR) < 5% serving as instrumental variables (Methods). A total of 5662 proteins with a cis-acting pQTL instrument, referred to as A-proteins (Fig. 1), were identified. For each A-protein, we calculated the posterior probability of it having a causal effect on the serum levels of each of the remaining 7523 proteins (Methods), referred to as B-proteins or targets, yielding approximately 42.5 million potential network edges. From all highly significant edges (FDR = 1%), we defined each network regulator and its targets as a subnetwork of the global CPN. At this stage, nearly half of all network regulators had only a single target and at more permissive FDR thresholds, the proportion of A-protein with just one target decreases further (Supplementary Fig. S2). Since our primary focus was on regulatory proteins that accounted for the most variation in the serum proteome, we selected A-proteins with a minimum of 10 targets, leading to a global network consisting of 43,528 edges and 234 A-proteins. For A-proteins with multiple aptamers, we selected those with the largest number of targets, and further refinement of LD among cis-acting instruments resulted in the final CPN comprising 185 A-proteins (referred to here as network regulators), their corresponding subnetworks, and a total of 31,358 edges (Supplementary Data 1 and Fig. 2a). We identified a high number of indirect regulations (Fig. 2a), where two network regulators ($x$ and $y$) are responsible for the regulation of a common set of targets ($z$), but $y$ is also a target of $x$. Such motifs are known as feed-forward loops (FFLs) and are a common feature in biological networks[29].

Independent cis instruments accounted for an average of 7.4% of the variance in protein levels across networks, with some cis signals contributing up to 84% of the variance (Methods). Furthermore, by utilizing cis signals for parental nodes (Methods, Supplementary Text), we observed a correlation between the number of regulators a target protein has, and the proportion of total variance in target protein expression explained by the cis signal of the network regulator (Spearman $r = 0.78$) (Supplementary Fig. S3). In some cases, more than 50% of the variance in the expression of the target protein was

**Table 1 | Baseline characteristics of the AGES study participants, stratified by incident myocardial infarction (MI)**

| Characteristic | Variable* | Free of MI | Incident MI | P-value | Total |
|---|---|---|---|---|---|
| **Demographics** | Numbers | 4051 (86.8) | 618 (13.2) | N/A | 4669 |
| | Females (%) | 2507 (61.9) | 299 (48.4) | 3E-10 | 2806 (60.1) |
| | Age (years) | 76.14 (5.46) | 78.64 (5.70) | 3E-24 | 76.44 (5.56) |
| **Anthropometry** | BMI (kg/m$^2$) | 27.05 (4.49) | 27.04 (4.22) | 0.975 | 27.05 (4.46) |
| | BMI category | – | – | 0.267 | – |
| | BMI < 25 kg/m$^2$ | 1371 (33.9) | 194 (31.4) | | 1565 (33.5) |
| | BMI = 20-30 kg/m$^2$ | 1769 (43.7) | 291 (47.2) | | 2087 (44.7) |
| | BMI > 30 kg/m$^2$ | 909 (22.4) | 132 (21.4) | | 1041 (22.3) |
| | SAT (cm$^2$, CT) | 260.50 (113.35) | 245.68 (104.99) | 0.003 | 258.85 (112.38) |
| | VAT (cm$^2$, CT) | 169.62 (78.65) | 180.19 (83.53) | 0.002 | 170.91 (79.38) |
| **Lifestyle** | Smoker | – | – | 0.039 | – |
| | Never | 1770 (44.8) | 236 (39.3) | | 2006 (43.0) |
| | Former | 1713 (43.3) | 282 (47.0) | | 1995 (42.7) |
| | Current | 472 (11.9) | 82 (13.7) | | 554 (11.9) |
| **Physiological** | DBP (mmHg) | 79.98 (10.71) | 80.64 (10.78) | 0.174 | 80.06 (10.71) |
| | SBP (mmHg) | 151.15 (22.29) | 155.94 (24.10) | 4E-06 | 151.76 (22.59) |
| | eGFR (ml/min/1.73m$^2$) | 65.19 (15.06) | 59.96 (16.18) | 2E-13 | 64.51 (15.30) |
| **Cardiovascular imaging** | CAC | 203.5 [26.9, 684.5] | 677.6 [195.5, 1609.1] | 3E-49 | 236.2 [34.8, 786.9] |
| | TAC | 202.5 [20.6, 826.7] | 556.9 [101.3, 1549.5] | 2E-20 | 229.8 [25.9, 906.4] |
| | Plaque severity | 2400 (64.0) | 450 (79.1) | 3E-13 | 2850 (61.0) |
| **Metabolic** | T2D | 432 (10.7) | 99 (16.0) | 2E-04 | 531 (11.4) |
| | MetS | 782 (19.3) | 144 (23.3) | 0.023 | 926 (19.8) |
| | HbA1c | 0.48 (0.09) | 0.50 (0.10) | 1E-04 | 0.49 (0.09) |
| | Incident NAFLD | 37 (5.6) | 9 (9.7) | 0.185 | 46 (0.98) |
| | HDLC (mmol/L) | 1.62 (0.45) | 1.53 (0.45) | 2E-07 | 1.61 (0.45) |
| | LDLC (mmol/L) | 3.56 (1.01) | 3.55 (1.07) | 0.848 | 3.56 (1.02) |
| | TG (mmol/L) | 1.03 [0.78, 1.41] | 1.08 [0.78, 1.53] | 0.019 | 1.04 [0.78, 1.42] |
| **Medication** | Lipid lowering | 699 (17.3) | 151 (24.4) | 3E-05 | 850 (18.2) |
| | Antihypertensive | 2406 (59.4) | 448 (72.5) | 3E-10 | 2854 (61.1) |
| **Cardiovascular and follow-up** | Incident CHD | 384 (9.5) | 476 (77.0) | 2E-274 | 860 (18.4) |
| | Incident HF | 188 (4.6) | 159 (25.7) | 3E-55 | 347 (7.4) |
| | Incident Stroke | 368 (9.1) | 54 (8.7) | 0.838 | 422 (9.0) |
| | Follow-up (years) | 9.9 [8.1, 11.0] | 5.6 [2.8, 8.2] | 6E-136 | 9.7 [6.8, 10.8] |

*Continuous variables were compared with two-sample t tests or Wilcoxon rank-sum tests, as appropriate, and categorical variables with the Chi-square test. Numbers are mean (SD) for continuous-, N (%) for categorical- and median [IQR] for skewed variables. All P-values reported are two-sided. The LDL cholesterol and blood pressure readings were adjusted for lipid-lowering and antihypertensive medications, respectively. Abbreviations: *BMI* body mass index, *SAT* subcutaneous fat by CT, *VAT* visceral fat by CT, *SBP* systolic blood pressure, *DBP* diastolic blood pressure, *HDLC* HDL cholesterol, *LDLC* LDL cholesterol, *TG* triglyceride, *FG* fasting blood glucose, *HbA1c* glycated hemoglobin, *T2D* type 2 diabetes, *MetS* metabolic syndrome, *NAFLD* non-alcoholic fatty liver disease, *CHD* coronary heart disease, *MI* myocardial infarction, *HF* heart failure, *CAC* coronary artery calcium, *TAC* thoracic aortic calcium; Plaque, carotid plaque severity (carotid plaque was assessed in 5017 individuals of the AGES cohort); *N/A* not applicable.
All participants were measured for 7523 proteins in serum. The table was generated after removing all prevalent MI cases.

explained solely by the *cis*-acting pQTLs of the parent proteins, with no contribution from local *cis*-components (Supplementary Text).

### High robustness and edge precision in the circulating causal protein network

We found the CPN to be robust in response to hub removal (Fig. 2b), with the largest connected component size still containing more than 80% of all nodes following the removal of the top 10 largest hubs, suggesting high levels of co-regulation and biological redundancy. The CPN also exhibited the typical "scale-free" property of biological networks[30], where a small number of regulators have a very high number of targets (Fig. 2c, d). We also examined the distribution of incoming edges and found that many of the network regulators with a high number of outgoing edges also have many incoming edges, further highlighting the interconnectedness of the global CPN (Fig. 2e, f).

Network robustness analysis and Precision-Recall assessments for the networks were conducted across various FDR thresholds and sample sizes. This involved random sampling of AGES individuals at varying sample sizes (Methods) and demonstrated a mean area under the receiver operating characteristic (ROC) curve exceeding 90%, even when the sample size was reduced to 2000 (Fig. 2g). However, when represented as a Precision-Recall curve, the precision at an FDR of 1% declined sharply to higher recall levels for sample sizes below 3000 and was even lower at more permissive FDR thresholds (Fig. 2h and Supplementary Fig. S4). Lastly, pairwise correlations between network targets were significantly stronger (Kruskal-Wallis *P*-value < $10^{-300}$) compared to randomly selected protein groups of the same size (Supplementary Fig. S5). In summary, these findings emphasize the strengths of the AGES study's large sample size, highlighting the robustness and accurate edge estimation of the identified protein network structure.

### Hierarchical organization of the circulating causal protein network

We used a greedy heuristic to derive a directed acyclic graph (DAG) from the 1622 interactions between network regulators (Methods,

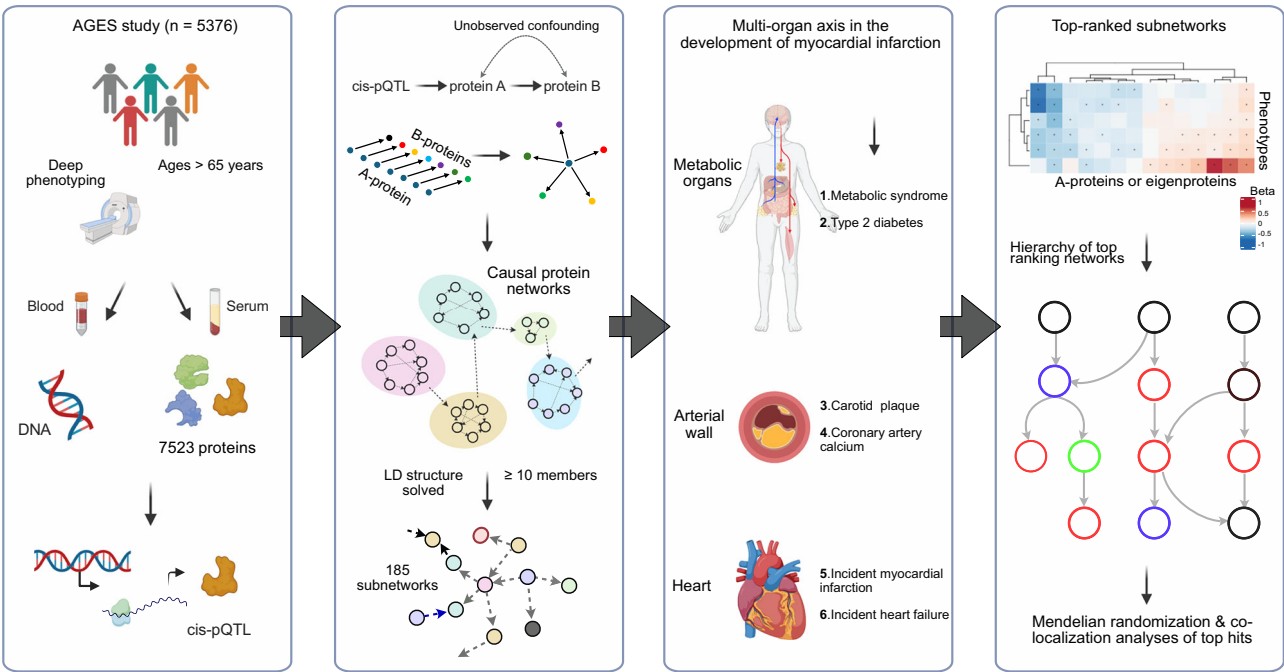

**Fig. 1 | A flowchart outlining the study overview and approach.** The schematic outlines the analytical pipeline, beginning (left) with the AGES prospective population-based cohort, which provided comprehensive phenotype and genotype information, proteomic profiles and *cis*-acting pQTL data. Subsequent steps included reconstructing 185 causal protein networks, each comprising 10 or more targets, *via* Mendelian randomization analysis, and evaluating associations between network regulators and eigenproteins with ACVD-related phenotypes, and finally (right), ranking CPN subnetworks by their connectivity to incident MI and related traits, revealing their hierarchical organization.

Fig. 2a), allowing for an interrogation of the hierarchical structure between said regulators. This process led to the removal of 572 edges between network regulators, resulting in an acyclic network with 1050 interactions, and the nodes then arranged at different levels according to their distance from the roots of the network (Fig. 3). We then reintroduced the removed edges to the CPN, while maintaining the ordered layout, and found much of the hierarchical structure to be conserved within the different levels (Supplementary Fig. S6). Most edges between proteins were observed within single levels, either above or below, with a few crossing multiple levels, indicating a highly structured organization within these networks. A hierarchical structure with a small number of levels has been observed in other biological networks[30,31].

Our approach infers causal relationships between protein pairs (A → B), whether directly or indirectly mediated by a third variable (e.g., A → C → B). The abundance of FFLs and within-level interactions may therefore reflect the high sensitivity of the AGES proteomics data for detecting weaker indirect causal effects. To address this, we generated a version of the directed acyclic graph (DAG) with all indirect edges removed, known as the transitive reduction of the DAG (Methods). This simplified, cascade-like network, comprising 255 edges, showed that many previously highly interconnected nodes remained prominent even after indirect edges were excluded (Supplementary Data 2).

Protein-protein interaction (PPI) networks are highly modular and feature connected hub proteins, reflecting a hierarchical biological organization[32]. Using the human integrated protein-protein interaction reference (HIPPIE)[33] database, we identified edges in the serum protein network overlapping with direct physical PPI networks, as outlined in the Supplementary Text. Specifically, across all confidence thresholds, we found that the true CPN captured significantly more PPI edges than the average observed in random networks, including the highest confidence PPIs (z-score = 24.3, *P*-value < 0.001) (Methods, Supplementary Text, Supplementary Fig. S7). This comparison highlights that a substantial portion of the CPN may reflect direct physical

interactions, most of which have been identified in vitro from solid tissues[33]. However, unlike the CPN, the PPI network database cannot capture interactions that span tissue boundaries within the context of the serum.

## Causal protein networks linked to ACVD related outcomes

We assessed the association of each network regulator and its corresponding CPN eigenprotein with incident MI and ACVD-related cardiometabolic traits, including MetS, T2D, CAC, carotid artery plaque burden, and incident HF (Methods). For this analysis, the first principal component (PC1) of each of the 185 subnetworks was calculated, and PC1s accounting for more than 15% of the variance within their respective subnetworks were designated as eigenproteins. Three CPN subnetworks did not meet this criterion and, therefore, did not have valid eigenproteins. Overall, 50 network regulators and 36 eigenproteins were significantly associated with incident MI (Supplementary Data 3, 4). Figure 4a–f illustrates the correlation between network regulators and their corresponding eigenproteins, highlighting their associations with various ACVD-related outcomes. For example, both the network regulator inter-alpha (globulin) inhibitor 3 (ITIH3) and the eigenprotein for that subnetwork were significantly associated with carotid plaque burden (Fig. 4c). When considering the network regulators, seven CPN subnetworks showed significant links to all six ACVD-related traits (Fig. 5a), while for the eigenproteins, only one CPN subnetwork (ITIH3) was associated with all six traits (Fig. 5b).

We ranked these subnetworks based on their associations with incident MI and other ACVD-relevant outcomes, assigning equal weight to both the network regulator and its corresponding eigenprotein (Supplementary Data 5, 6). This approach identified 25 top-ranked subnetworks (arbitrary rank score ≥ 7; Table 2). Clustering the network regulators based on their associations with these outcomes revealed that the six core traits align with the three established etiological axes of MI (Fig. 5c), as depicted in Fig. 1. In contrast, the

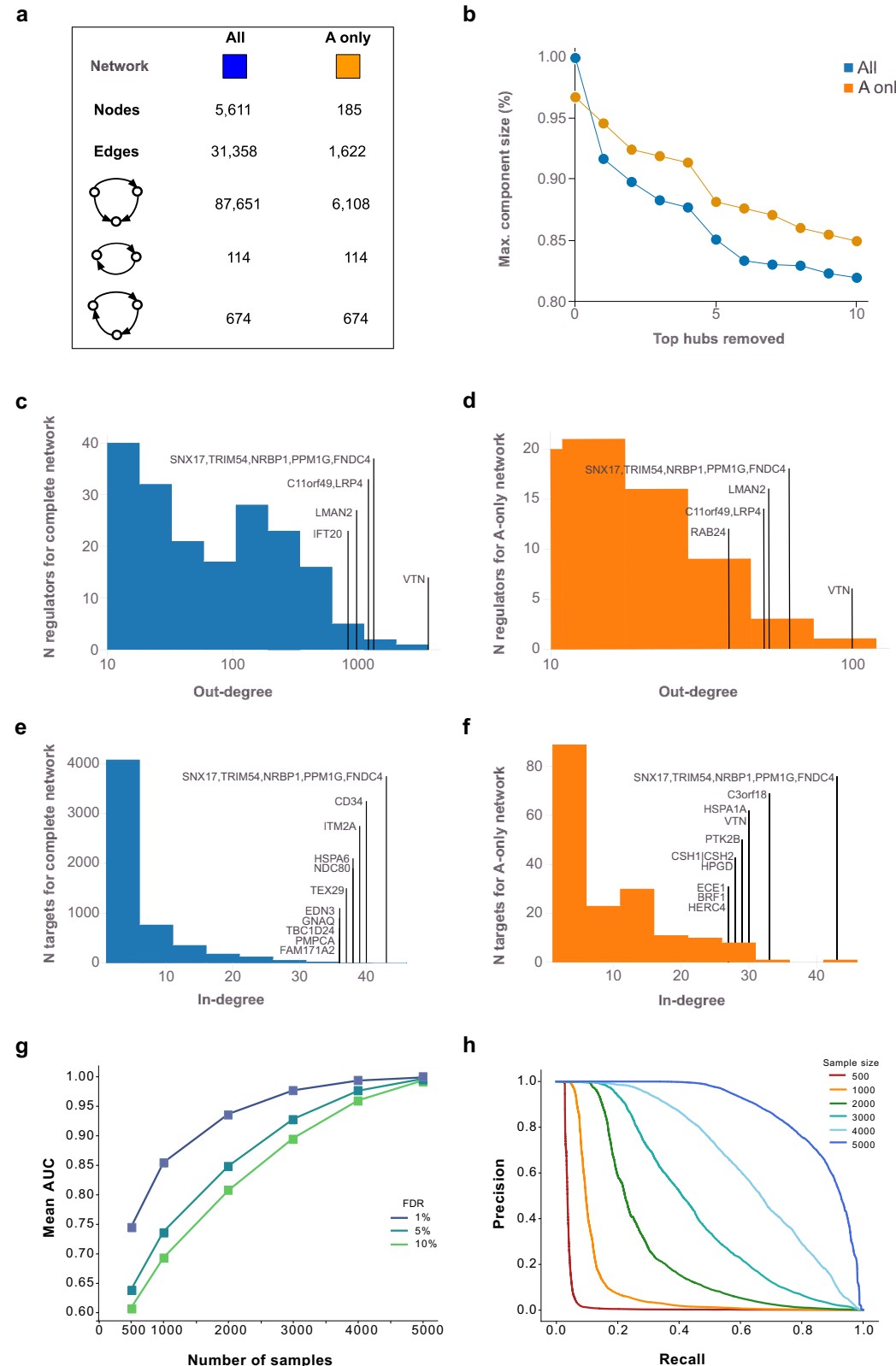

corresponding eigenprotein associations drive a distinct clustering of phenotypes, grouping CAC, carotid plaque burden, and MI together (Fig. 5d), which represent the key outcomes associated with atherosclerosis[34]. The top-ranked CPN subnetworks in Table 2 were largely retained when a stricter 30% eigenprotein variance threshold was applied (Supplementary Data 7, 8). Associations of all top ranked network regulators with incident MI and/or related traits are illustrated

in Supplementary Fig. S8. We note that a substantial number of the 182 subnetworks, including 62 network regulators and 121 eigenproteins, showed only one or no significant associations with ACVD-related traits (Supplementary Data 5, 6). In summary, both the top ranked network regulators and their corresponding eigenproteins capture the established multi-dimensional axes of ACVD development, albeit through slightly different mechanisms.

**Fig. 2 | Analysis of degree distribution and robustness in causal protein networks. a** The figure summarizes key statistics for both the full causal protein network (CPN; "all") and the subset restricted to edges among the 185 causal protein network regulators with at least 10 members ("A-only"). Reported metrics include the numbers of nodes and edges, as well as counts of 2- and 3-node feedback and feedforward loops. **b** The figure displays the relative sizes of the largest connected component after sequentially removing the top $k = 1$ to 10 hub proteins. **c** Out-degree distributions for network regulators are presented for the complete

network (blue) and (**d**) the A-only network (orange), with annotations emphasizing the most highly connected network regulators. **e** The corresponding in-degree distribution is shown for the network targets for the complete network (blue) and (**f**) A-only network (orange). **g** Robustness analysis receiver operating characteristic (ROC) area under the curve (AUC) for sub-sampled networks compared to networks using all 5376 samples. **h** Precision-Recall curves for sub-sampled networks at a 1% FDR threshold are compared to networks using all samples. Similar curves at 5% and 10% FDR thresholds are presented in Supplementary Fig. S4.

## The top ranked subnetworks exhibit a strong degree of interconnectivity

Having identified subnetworks of the global CPN that are associated with different aspects of ACVD, we were interested in comparing structural similarities between these subnetworks. Among the 25 top ranked subnetworks (Table 2), we find examples of clusters of proteins that are co-regulated by several network regulators (Fig. 6a). Furthermore, target similarity analysis identified clusters of CPN subnetworks that are both correlated through their eigenprotein and share similar targets (Fig. 6b). Interestingly, we find a large group of negatively correlated subnetworks which share similar targets, indicating co-regulation by distinct network regulators with opposing directional effects.

Like the global CPN, we observed multiple levels of regulation among the network regulators of the top ranked networks (Fig. 6c). The connected subnetworks form distinct modules that largely preserve this structure, even when the previously removed edges are reintroduced (Supplementary Data 1). This includes, for example, a cascade involving the network regulators C2, CFB, and GAL3ST1 (Fig. 6c), as well as the CFHR1 network regulator when applying a more stringent eigenprotein variance threshold (Supplementary Fig. S9). This is noteworthy because CFHR1, C2 and CFB are all integral components of the same complement cascade, whereas GAL3ST1 is an enzyme involved in the biosynthesis of glycosaminoglycans, which play vital roles in various biological processes, including inflammation and vascular remodeling[35]. The role of the complement system in inflammatory processes and cardiac health is well-established, particularly in relation to myocardial tissue injury[36]. Another intriguing example involves a pathway of five proteins: COL28A1, NUDT21, PTPN11, HSPA1B, and HSPA1A (Fig. 6c). Notably, all but COL28A1 are enriched in the hematopoiesis pathway (g:Profiler, P-value = 0.018), essential for blood cell formation and development from hematopoietic stem cells. This process is critical for maintaining homeostasis and has broad implications in atherosclerosis[37]. Interestingly, mutations in the mouse *Col28a1* gene are linked to abnormal blood vessel morphology, according to the MGI database[38]. These cascade-related proteins may collectively influence inflammation, vascular remodeling, atherosclerosis, and blood vessel integrity, supporting their observed associations with incident MI and related traits in this study.

Finally, comparison of the top ranked CPN subnetworks with the previously published serum protein co-regulatory network from the AGES study[23], revealed substantial overlap between protein clusters in both network types (Supplementary Data 9, Supplementary Fig. S10, Supplementary Text). This overlap is evident in two ways: first, many CPNs align with the same co-regulatory module; second, when a single CPN intersects with multiple co-regulatory modules, these modules often belong to the same super-cluster of correlated co-regulatory modules (Supplementary Text). Overall, a significant relationship exists between the circulating CPNs and the co-regulatory networks, despite fundamental differences in the methodologies used to reconstruct them.

## Replication of causal protein network architecture in an independent cohort

We performed a validation analysis of the CPN structure using the UK Biobank (UKBB) study[39] as an independent dataset. There are 2186

assays from the UKBB Olink Explore and the AGES 7K SOMAscan profiling platforms that target the same proteins according to their annotations (Methods). Therefore, we reconstructed CPNs in both AGES and UKBB using only this subset of proteins from each platform to ensure that these networks were comparable. As correlations in protein measurements between these two different proteomic platforms have been shown to vary[40], we conducted instrument selection separately in the UKBB and AGES studies (Methods), removing duplicate aptamers targeting the same protein before analysis.

We first reconstructed a subsetted CPN in AGES but restricted it to the 2186 proteins also measured in UKBB. Of these protein assays, 1835 had a valid *cis*-acting pQTL instrument to be used as a potential network regulator. At a 1% FDR threshold, we identified 454 proteins with at least one target and 43 with at least 10 targets in the AGES study. Among the 454 proteins with at least one target in the subsetted AGES CPN (FDR = 1%), 416 were also present in the full AGES CPN constructed from all measured proteins. Furthermore, 96.3% of the edges identified in the subsetted AGES CPN overlapped with those in the full CPN. We then reconstructed a comparative CPN using data from 52,543 UKBB participants with Olink assays targeting the same 2186 proteins. We observed a significant overlap between network regulators in the two cohorts, including the 454 proteins with at least one target in AGES, of which 393 were present with at least one target in UK Biobank (Fisher's Exact Test; OR = 1.9, $P = 4.1 \times 10^{-6}$). Further, we identified 629 edges shared between the AGES subsetted and UK Biobank CPNs (FDR = 1%). To formalize this comparison, we applied Fisher's Exact Test, which demonstrated that the overlap was highly significant (OR = 2.9, $P = 1.6 \times 10^{-99}$), using all possible edges as the background while excluding self-edges. Notably, many more targets were identified for a given CPN in UKBB than in AGES (Supplementary Data 10), likely to reflect the nearly tenfold larger sample size in UKBB.

We next sought to determine the extent to which protein targets of subnetworks reconstructed in AGES and those from UKBB overlapped. Of the 393 shared network regulators from the comparative subnetworks, we found 36 that shared at least 2 targets in the AGES and UKBB comparative CPNs, for which we assessed overlap significance with a hypergeometric test, using the full set of 2186 proteins as background. Overall, 23 subnetworks had a significant overlap in targets across both platforms ($q < 0.05$), including three of the top-ranked subnetworks previously linked to ACVD in the full AGES CPN, namely, KLKB1, CFB and CSF3 (Supplementary Data 10 and Supplementary Fig. S11).

A recent study compared correlations between Olink and SOMAscan platforms in the Icelandic population[40]. From this study, we obtained correlations for 1677 of the 2186 proteins measured (mean Pearson coefficient = 0.39). For 19 of the 23 significantly overlapping subnetworks, correlations were available, with a higher mean of 0.48 (Supplementary Data 10). In some cases, network regulators of significantly overlapping CPNs were not correlated across platforms (Supplementary Data 10), suggesting that even when different affinity-based technologies bind distinct epitopes or proteoforms, the network regulator remains robust, continuing to capture the same underlying biology reflected in its downstream targets.

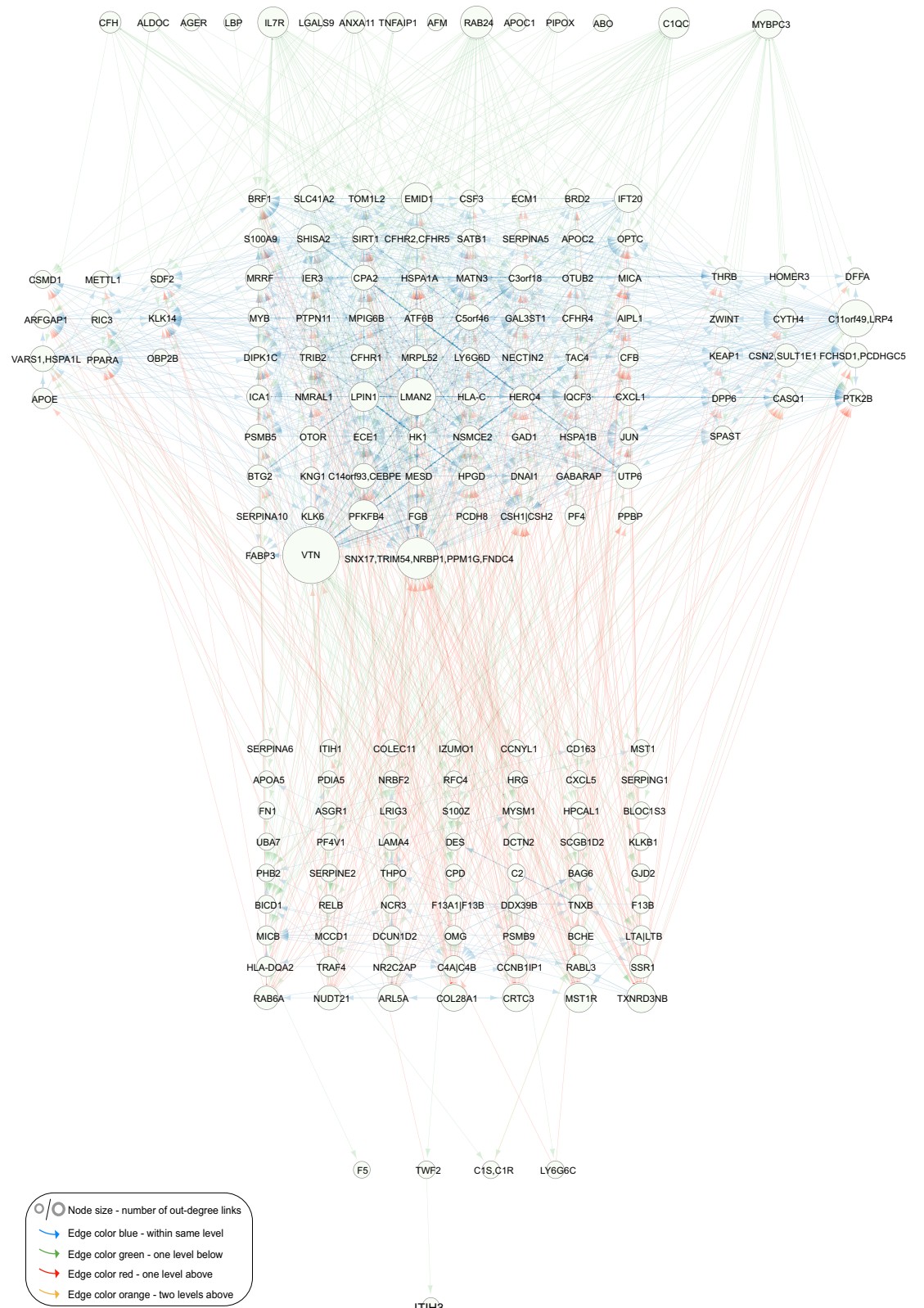

**Fig. 3 | Interactions between network regulators of the circulating causal protein network.** The network regulator only directed acyclic graph (DAG) of the circulating causal protein network (CPN). Network visualization of causal interactions between the network regulators of the 185 subnetworks with 10 or more targets (FDR = 1%). Node size is proportional to the number of out-degree links. The edges are color-coded according to levels (hops): blue for connections within the same level, green for links to one level below, red for links to one level above, and orange for links to two levels above. A similar network visualization is presented in Supplementary Fig. S6, where no edges have been removed.

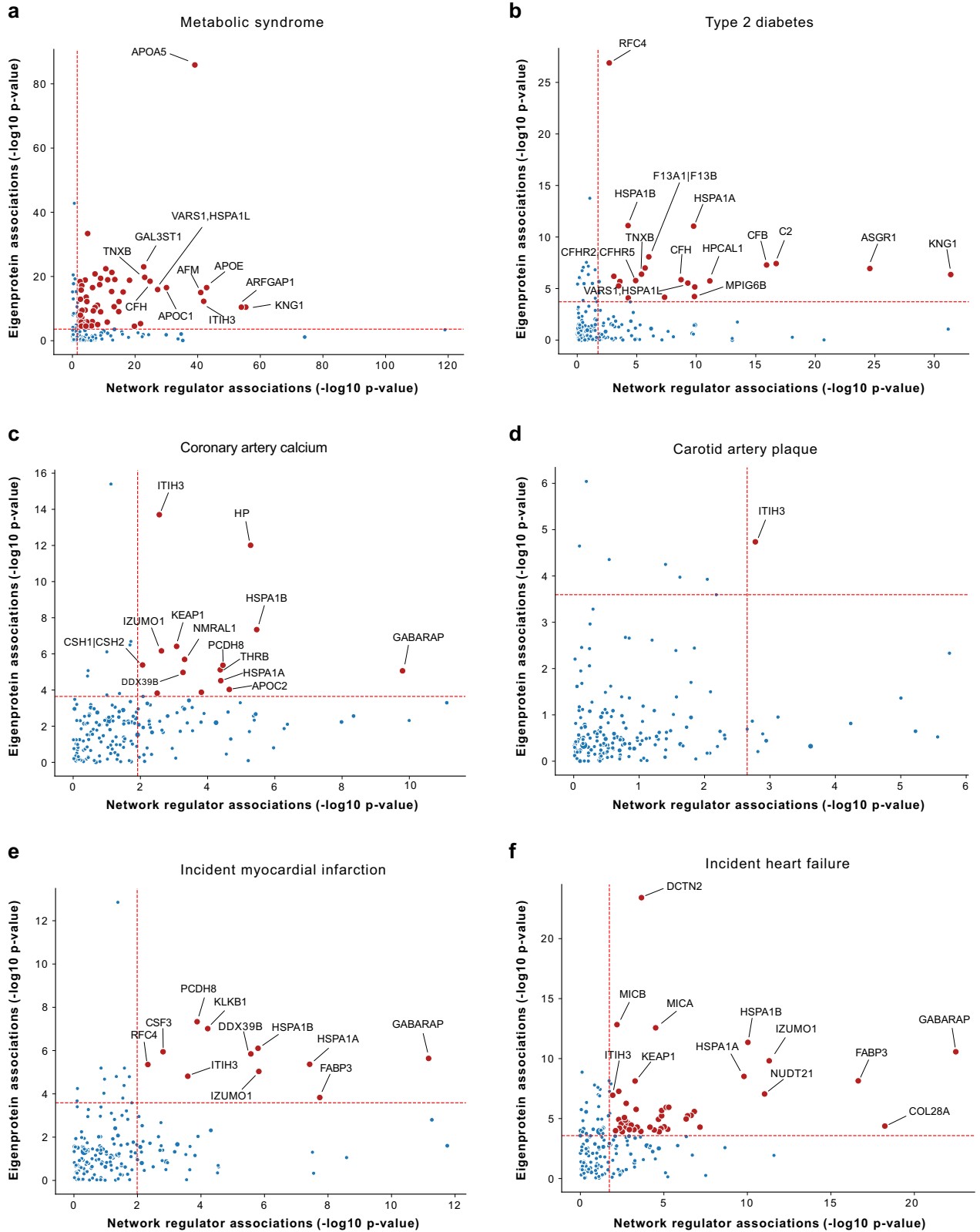

**Fig. 4 | Comparison of association *P*-values for regulators and eigenproteins with myocardial infarction and related traits.** Scatterplots (**a**–**f**) display the correlation of association *P*-values between 182 network regulators and their corresponding network eigenproteins (first principal component, explaining >15% of variance) for myocardial infarction (MI) and cardiometabolic traits contributing to atherosclerotic cardiovascular disease (ACVD), represented as -log₁₀(*P*-values). The relationship between serum protein (and eigenprotein) levels and quantitative phenotypes was evaluated using linear regression controlling for age and sex, whilst the relationship between serum proteins (and eigenproteins) and prevalent disease was examined cross-sectionally using logistic regression with age and sex adjustments. The associations between serum proteins (and eigenproteins) and incident disease were evaluated longitudinally using the Cox proportional-hazards model. Red dashed lines mark the 5% FDR thresholds for the respective axes.

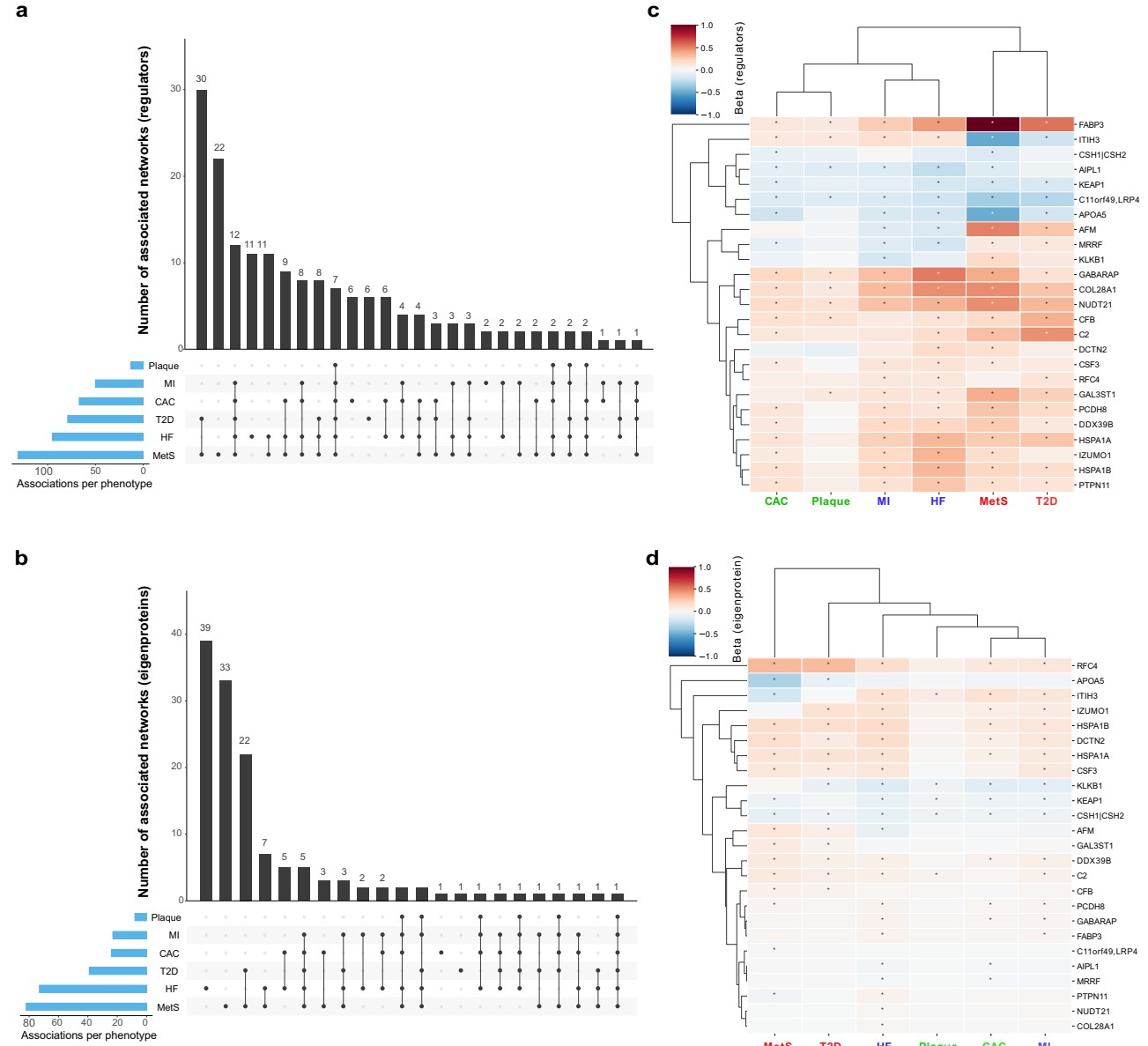

**Fig. 5 | Intersection and clustering of subnetworks linked to myocardial infarction and related traits.** The UpSet plots illustrate the intersections of the 182 causal protein networks (CPN) related to incident myocardial infarction (MI) and related cardiometabolic trait associations with (**a**) the network regulators (FDR ≤ 5%) and (**b**) the CPN eigenproteins (FDR ≤ 5%). The heatmaps on the right depict the relationship between the top-ranked subnetworks (**c**) network regulators and (**d**) eigenproteins and MI-related outcomes. Blue squares indicate a negative relationship between network regulators or eigenproteins and the outcome, while red squares represent a positive relationship. A star within a box denotes a significant association between proteins and outcome, based on an FDR estimate of < 5%. CAC coronary artery calcium, Plaque carotid artery plaque burden, T2D type 2 diabetes (prevalent), MetS metabolic syndrome, MI incident myocardial infarction, HF incident heart failure.

## Additional links between the top ranked networks and ACVD

We investigated several complementary lines of evidence to support the role of the top-ranked network regulators in the pathophysiology of MI and related cardiometabolic traits. Network regulators and targets from the top-ranked subnetworks showed enrichment in multiple pathways previously linked to ACVD, including a shared enrichment in the cellular response to heat shock (Supplementary Data 11, Supplementary Figs. S12–S13, Supplementary Text). STRING-based analysis revealed significantly ($P$-value = 0.047) enriched functional and physical interactions among the top network regulators (Supplementary Fig. S14), including previously found (Fig. 6c) and previously unreported connections (e.g., ITIH3 with KLKB1, APOA5, and AFM; Supplementary Fig. S14).

**Causal inference analysis of top-ranked regulators.** Two-sample MR and colocalization analyses highlighted several top-ranked regulators, including APOA5, DDX39B, HSPA1A, C11orf49 and LRP4, with significant support for causal associations with ACVD-related traits (FDR < 0.05; Supplementary Data 12–13). Here, APOA5 demonstrated a strong protective effect against MI and MetS, consistent with findings that APOA5 knockout mice display a fourfold elevation in plasma triglyceride levels[41], a significant risk factor for ACVD[42]. Further, genetically determined levels of HSPA1A showed an association with MetS, indicating a causal role, whereas DDX39B was linked to both MetS and T2D. Notably, the HSPA1A gene has also been causally implicated in T2D and its microvascular complications in prior MR and colocalization analyses[43]. Sensitivity analyses, restricted to exposures with at

**Table 2 | The top ranking (score ≥ 7) causal protein networks related to incident MI and associated traits**

| Protein regulatory network | Network versus MI-related trait relationships* | | | | | | | Score |
|---|---|---|---|---|---|---|---|---|
| Network regulator | Number of B-proteins | CAC | Plaque severity | MetS | T2D | MI | HF | |
| ITIH3 | 11 | 2 | 2 | 2 | 1 | 2 | 2 | 11 |
| HSPA1B | 13 | 2 | 0 | 2 | 2 | 2 | 2 | 10 |
| DDX39B | 36 | 2 | 0 | 2 | 2 | 2 | 2 | 10 |
| HSPA1A | 19 | 2 | 0 | 2 | 2 | 2 | 2 | 10 |
| KEAP1 | 12 | 2 | 1 | 2 | 1 | 1 | 2 | 9 |
| C2 | 32 | 1 | 1 | 2 | 2 | 1 | 2 | 9 |
| PCDH8 | 131 | 2 | 0 | 2 | 1 | 2 | 2 | 9 |
| GABARAP | 143 | 2 | 1 | 1 | 1 | 2 | 2 | 9 |
| CSH1 \| CSH2 | 19 | 2 | 1 | 2 | 1 | 1 | 1 | 8 |
| RFC4 | 13 | 1 | 0 | 1 | 2 | 2 | 2 | 8 |
| IZUMO1 | 24 | 2 | 0 | 1 | 1 | 2 | 2 | 8 |
| CSF3 | 10 | 1 | 0 | 2 | 1 | 2 | 2 | 8 |
| FABP3 | 98 | 1 | 1 | 1 | 1 | 2 | 2 | 8 |
| KLKB1 | 18 | 1 | 1 | 1 | 1 | 2 | 1 | 7 |
| DCTN2 | 19 | 1 | 0 | 2 | 1 | 1 | 2 | 7 |
| AFM | 19 | 0 | 0 | 2 | 2 | 1 | 2 | 7 |
| AIPL1 | 138 | 2 | 1 | 1 | 0 | 1 | 2 | 7 |
| MRRF | 251 | 2 | 0 | 1 | 1 | 1 | 2 | 7 |
| APOA5 | 19 | 1 | 0 | 2 | 2 | 1 | 1 | 7 |
| PTPN11 | 83 | 1 | 0 | 2 | 1 | 1 | 2 | 7 |
| CFB | 97 | 1 | 1 | 2 | 2 | 0 | 1 | 7 |
| GAL3ST1 | 42 | 0 | 1 | 2 | 2 | 1 | 1 | 7 |
| NUDT21 | 396 | 1 | 1 | 1 | 1 | 1 | 2 | 7 |
| COL28A1 | 478 | 1 | 1 | 1 | 1 | 1 | 2 | 7 |
| C11orf49-LRP4 | 1208 | 1 | 1 | 2 | 1 | 1 | 1 | 7 |

*Linear, logistic, or Cox regression analysis was used depending on whether the traits were continuous, prevalent categorical, or incident categorical, respectively. Scoring: neither network regulator nor eigenprotein associated with the trait = 0; network regulator or eigenprotein (PC1 ≥ 15% variance explained) associated with the trait = 1; both network regulator and eigenprotein linked to the trait = 2. Statistical significance for network regulators was determined using the Benjamini-Hochberg adjusted FDR with a *P*-value threshold of < 0.05. For eigenprotein associations with traits, a Bonferroni-adjusted *P*-value threshold of < 0.00027 was applied.

*CAC* coronary artery calcium, Plaque severity, carotid plaque severity score; *MetS* metabolic syndrome, *T2D* type 2 diabetes (prevalent), *MI* myocardial infarction (incident), *HF* heart failure (incident).

least three instruments (Methods), indicated that results were largely robust across approaches (Supplementary Data 14), although significant instrument heterogeneity was observed for APOA5 in relation to MetS.

**Additional context from current and prior evidence.** Beyond the analyses of the present study, several top regulators, including ITIH3, HSPA1B, KEAP1, HSPA1A, C2, CSF3, FABP3, KLKB1, AFM, APOA5, PTPN11, CFB, and COL28A1, have previously been linked to ACVD or related cardiometabolic traits (Supplementary Text). For example, the top-ranked regulator ITIH3 has a gain-of-function genetic association with increased risk of MI and is expressed in vascular smooth muscle cells and macrophages within atherosclerotic plaques[44]. Notably, ITIH3 is positioned at the base of the data-driven CPN (Fig. 3), indicating a potential role in numerous regulatory pathways. Another example is KLKB1, which we recently demonstrated through MR analysis to be causally protective against calcific aortic valve disease[45], a leading cause of heart failure via aortic stenosis[46]. To further characterize the top-ranked protein network regulators, we assessed their effect directions on ACVD-related traits and survival outcomes (Supplementary Data 15) and evaluated if they are druggable or targeted by available compounds using public databases (Supplementary Data 16). Table 3 summarizes the accumulated findings, offering an integrated overview of directional consistency across causal, observational, and therapeutic evidence for each regulator.

Many of the top-ranked regulators showed consistent directionality across multiple lines of evidence and exhibit positive and often causal effects on ACVD traits (Table 3), including ITIH3, HSPA1A, DDX39B, FABP3, PTPN11, CFB, and COL28A1, whereas APOA5 and KLKB1 show protective effects (Table 3 and Supplementary Text). In contrast, network regulators such as KEAP1, C2, and CSF3 show inconsistent effect directions between the present findings and previously reported associations (Table 3). KEAP1 regulates the NRF2 pathway, essential for oxidative balance and cardiovascular protection (Supplementary Text). Cardiomyocyte-specific *Keap1* knockout upregulates NRF2 and prevents induced cardiac dysfunction[47], contrasting with the protective association of high circulating levels of KEAP1 in our study (Table 3). This discrepancy may reflect differences between tissue-specific and circulating biology, where serum KEAP1 could indicate systemic processes independent of NRF2 inhibition or a feedback marker of activated antioxidant responses rather than a direct protective factor. A human study found C2 deficiency to be associated with increased atherosclerosis risk[48], suggesting a protective role for C2 (Supplementary Text). In contrast, our finding that higher circulating C2 predicts increased incident MI risk (Table 3), may reflect differences between lifelong deficiency and elevated plasma levels, with the latter potentially indicating complement activation or inflammation. Alternatively, elevated C2 in the present study could represent a compensatory upregulation in response to subclinical vascular injury. Further, cardiac myocytes express CSF3 under both

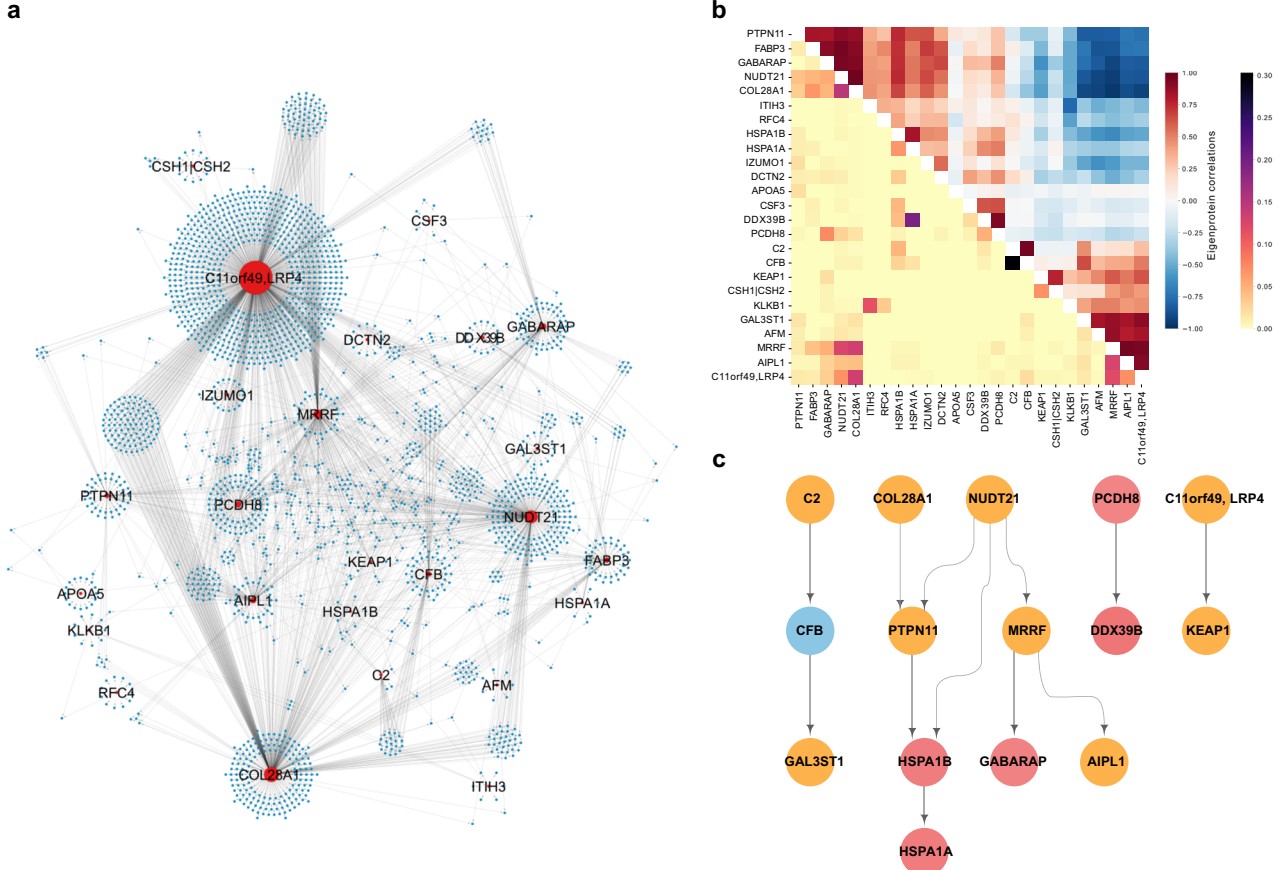

**Fig. 6 | Interconnectivity between the top ranked causal protein networks associated with myocardial infarction and/or related traits. a** Network visualization of the interconnections among the top ranked causal protein networks (CPN) for incident myocardial infarction (MI) and associated traits, where the variance explained by network eigenproteins is over 15% (see Table 2). Red nodes represent network regulators, while blue nodes denote target proteins. **b** Heatmap displaying the similarity between CPN subnetworks. The upper right triangle shows hierarchical clustering based on pairwise Pearson correlations between network eigenproteins. The clustering leaf order of the Pearson correlations has been applied to the lower left triangle, which represents pairwise Jaccard Index scores between the same networks. **c** Hierarchical representation of interacting network regulators where color indicates degree of association with incident MI. Red = both eigenprotein and network regulator are significantly (FDR < 0.05) linked to incident MI, orange = either eigenprotein or network regulator associated with MI, and blue = no association to incident MI.

normal and stress conditions, particularly following ischemic injury, where it supports cardiomyocyte survival after a heart attack[49]. Notably, the positive association between circulating CSF3 levels, incident MI, and reduced survival (Table 3), may reflect a compensatory or adaptive response to early cardiac injury or stress during subclinical disease (Supplementary Text). Finally, given the conflicting findings for C11orf49 and LRP4 in our study and the inconsistent evidence for AFM in previous research (Table 3), the direction of their effects on ACVD etiology remains inconclusive. In summary, 68% (17 of 25) of the top-ranked network regulators, this study, along with others, reveals additional specific connections to ACVD-related traits. These findings strengthen evidence for the causal role of many of the network regulators and their targets in ACVD development and may guide therapeutic strategies to modulate them.

**Therapeutic pathways and clinical implications.** Over half of the top-ranked network regulators are potentially druggable with small molecules or biologics, although only a few have compounds currently available for either related or unrelated disease indications (Supplementary Data 16, Supplementary Text). Specifically, 13 of the 25 regulators are druggable, with 6 already targeted by licensed or investigational drugs. Building on this, the CPN analysis has identified both interconnected networks and therapeutic pathways with potential for early ACVD detection and prevention: (1) Complement-

mediated inflammation: The network regulators C2 and CFB are central to the alternative complement pathway, driving vascular inflammation and atherosclerosis[50]. CFB is targeted by the antisense oligonucleotide Sefaxersen, in development for IgA nephropathy (Supplementary Data 16). Targeting this pathway could prevent early inflammatory progression of atherosclerotic lesions. (2) Cellular stress response and cyto-protection: KEAP1 regulates the Nrf2 oxidative stress pathway[47], while HSPA1A and HSPA1B control protein folding under stress[51]. All three have licensed drugs targeting them (Supplementary Data 16). Modulating this pathway may protect against oxidative damage and endothelial dysfunction. (3) Hemostasis: The key target KLKB1, a regulator of plasma kallikrein in the intrinsic coagulation pathway[52], is targeted by the approved drug Berotralstat against angioedema (Supplementary Data 16). Our integrative analysis, however, implies that activating KLKB1 may offer a strategy to prevent thrombotic complications in ACVD (Table 3). 4) Lipid metabolism: The network regulators APOA5 and AFM control triglyceride-rich lipoprotein metabolism and oxidative stress[41,53,54], are bio-druggable (Supplementary Data 16), and could be targeted to prevent dyslipidemia-driven initiation of atherosclerosis. This network framework highlights causal pathways for ACVD prevention, with concordant data strengthening target confidence and inconsistencies pointing to opportunities for novel therapeutic discovery.

**Table 3 | Summary of network regulators, including findings from the present study, previously known biological links, effect directions, if druggable or status as a target of an available compound**

| Regulator | Evidence from MR/Colocalization in AGES, and prior known biology related to ACVD* | Effect direction on ACVD and/or survival (Supplementary Data 15) | Druggable / available compound (Supplementary Data 16) |
|---|---|---|---|
| ITIH3 | Gain-of-function variant increases risk of MI. Highly expressed in atherosclerotic lesion | Risk | Druggable |
| HSPA1B | Causally linked to T2D and microvascular complications | Risk | Available compound |
| DDX39B | Causally related to T2D and MetS (AGES) | Risk | Not druggable |
| HSPA1A | Causally related to MetS (AGES). Causally linked to T2D and microvascular complications | Risk | Available compound |
| KEAP1 | Cardiomyocyte-specific knockout mice show improved cardiac function | Protective | Available compound |
| C2 | Deficiency increases atherosclerosis | Risk | Druggable |
| PCDH8 | None | Risk | Not druggable |
| GABARAP | Genetic links to hypertension and CVD | Risk | Not druggable |
| CSH1/CSH2 | None | Protective | Druggable |
| RFC4 | Genetic links to diabetes and venous thromboembolism | Risk | Not druggable |
| IZUMO1 | None | Risk | Not druggable |
| CSF3 | Promotes cardiomyocyte survival after MI | Risk | Druggable |
| FABP3 | Contributes to ischemic heart injury | Risk | Not druggable |
| KLKB1 | Causally protective against calcific aortic valve disease; influences cholesterol efflux | Protective | Available compound |
| DCTN2 | None | Risk | Not druggable |
| AFM | Positively associated with the metabolic syndrome, but lower levels in MI patients | Protective | Druggable |
| AIPL1 | None | Protective | Not druggable |
| MRRF | None | Protective | Not druggable |
| APOA5 | Causally protective against MI and MetS (AGES). Knock-out leads to elevated plasma triglycerides | Protective | Druggable |
| PTPN11 | Gain-of-function mutations cause heart defects in oonan syndrome | Risk | Available compound |
| CFB | Knock-out mice show improved metabolic and cardio-vascular features | Risk | Available compound |
| GAL3ST1 | None | Risk | Not druggable |
| NUDT21 | None | Risk | Not druggable |
| COL28A1 | Positively associated with HF and its subtypes | Risk | Druggable |
| C11orf49 | Causally linked to MI (AGES) | Protective | Not druggable |
| LRP4 | Causally linked to MI (AGES) | Inconclusive | Not druggable |

*Background on previously characterized biological roles of network regulators: ITIH3[44]; HSPA1A/B[43]; KEAP1[47]; C2[48]; GABARAP[94]; RFC4[94]; CSF3[49]; FABP3[95]; KLKB1[45,96]; AFM[53,54]; APOA5[41]; PTPN11[97];CFB[98]; COL28A1[99]. The Supplementary Text summarizes prior literature on the roles of many network regulators in ACVD-related etiology.

## Discussion

Atherosclerotic cardiovascular disease remains the leading cause of death worldwide[1], emphasizing the urgent need for early detection and innovative preventive strategies. Clinical complications to ACVD like MI, stroke, and its longer-term complication HF, arise from a complex interaction of both local (cardiovascular) and systemic (non-cardiovascular) factors, highlighting the importance of a broad systems-level approach to understand their intricate etiology. To address this, we employed a causal inference approach to reconstruct the first large-scale circulating CPN in humans. We identified 185 CPN subnetworks, many of which were strongly associated with the onset of MI and/or HF, as well as with upstream clinical risk factors essential to ACVD development. The strongest CPN subnetworks associations clustered these phenotypes according to their localization within the key organ axes central to ACVD related pathophysiology. This study begins to uncover the underlying systemic mechanisms, reflected in serum proteins, that drive processes leading up to the onset of MI and its long-term consequence, HF.

The central dogma of molecular biology outlines the process of information transfer from DNA to proteins. Proteins have the

capability to influence phenotypes, including disease, through biological networks. The capacity to acquire measurements of thousands of proteins from a single blood sample has opened new paths for monitoring health and disease in greater depth than ever before[27,55]. We identified the first circulating protein co-regulatory networks in humans, linking it to the genome in an unbiased manner and elaborating on those findings to highlight its connections to complex disease[23]. Despite being highly informative, the previously described co-regulatory network[23] is undirected and subject to confounding, which means that the causal relationship between nodes is unknown. More recently, large-scale directed single-tissue and cross-tissue gene-regulatory co-expression networks have been reconstructed across multiple tissues related to CAD, providing a comprehensive mechanistic framework for understanding the etiology of the disease[11]. Causal inference and causal networks present an ideal framework for graphically modeling complex systems because it allows for the transmission of prior information about a system and the formulation of concrete hypotheses for follow-up research[56]. As such, causal inference is situated between purely data-driven machine learning and detailed mechanistic modeling approaches. Causal network inference,

however, is a computationally demanding and complex process, often restricted to small-scale systems. To overcome this, we developed a new approach that outperforms previous methods in both efficiency and coverage, and proving particularly effective when both genotype and molecular node data are available for the same sample[19,20]. In this study, we used this approach within the single-center population-based AGES study. Each participant provided essential genotype, proteomic, and clinical data, which allowed for the identification of the circulating CPN at a low FDR threshold, as well as the detection of subnetworks associated with the future onset of ACVD.

Although circulating co-regulatory networks and CPNs are reconstructed using different methodologies, we observed a significant overlap between these two types of networks. CPNs comprise a single regulator and its target protein nodes, while co-regulatory networks encompass a much broader scope. In fact, because many CPNs interact and often overlap within the same co-regulatory module, they may converge to create a larger co-regulatory module. The significant overlap between the two types of networks is not entirely unexpected, as co-regulatory serum protein networks have been shown to be strongly influenced by genetic factors[23], and proximal pQTL were employed as instrumental variables in reconstructing the CPN. This suggests that the overlapping protein clusters shared between the two network types are likely influenced by the same genetic factors. However, CPNs offer additional insights for causal relationships between proteins that are not evident in co-regulatory networks. In other words, the CPN can illustrate how changes in one protein node can regulate another node even in the presence of unobserved confounding factors, effectively distinguishing between the effects of correlation and causation. This ultimately improves the modeling of complex disease etiology.

Validating our CPNs is inherently challenging because complementary external data are lacking in depth, unlike tissue-specific gene regulatory networks that can be supported by gene expression QTLs and external transcription factor-target datasets[57]. To address this challenge, we drew on existing resources such as the UKBB[58,59], although its protein coverage is considerably smaller than in our study, and inter-platform correlations are modest (mean $r = 0.39$) with additional differences arising from the epitope targeting inherent to affinity-based technologies. Nevertheless, the identification of 23 significantly overlapping subnetworks (out of 36 comparable networks) across AGES and UKBB highlights that the CPN architecture is robust. We further observed strong cross-cohort replication of global regulators and their connections, with regulators and edges significantly overlapping beyond chance expectations. The UKBB's tenfold larger sample size facilitated the detection of more targets per regulator, but its reduced proteomic coverage limited a more comprehensive replication of networks, emphasizing the dual importance of sample size and proteomic breadth for network reconstruction and comparison. Importantly, several of the top-ranked subnetworks, including KLKB1, CFB, and CSF3, were consistently replicated across platforms and populations, underscoring their biological relevance beyond platform-specific effects.

A key strength of our study is the high-quality data from a large-scale, prospective, population-based cohort, which includes detailed phenotype information for each participant, extensive coverage of circulating proteins, and comprehensive genomics data. This study, however, has several limitations that must be acknowledged. First, all AGES participants are of European descent (i.e., white/Caucasian), which may limit the generalizability of our findings, as protein predictors and clinical indicators of heart disease can differ across populations with diverse genetic and environmental backgrounds. In addition, the present findings are based on serum proteins and may not fully capture the MI-related pathobiology in solid tissues, such as the arterial wall and heart. Moreover, the study does not encompass the entire serum proteome, which is still being mapped; however, it

remains one of the largest studies of its kind to date. Regarding the CPNs, we still do not have a clear understanding of the nature of the edges in our serum protein networks, especially whether they represent direct or indirect regulation mediated by processes within or across tissue boundaries. Our findings, however, indicate that while the CPNs partially reflect protein-protein interactions, they may also capture interactions between nodes across tissues. In addition, a high number of indirect interactions have been observed in these networks; however, it is not currently possible to determine which of these are true instances of FFLs, commonly enriched in biological networks, or false positives. Developing causal methods that use higher-order mediation tests between triplets of proteins could address this issue. Current mediation tests, however, suffer from hidden confounding, which is accounted for within randomization-based approaches.

Our integrative analyses support a key mechanistic role for many top-ranked network regulators in ACVD, with mostly consistent associations observed across complementary evidence. Some regulators, including KEAP1, C2, and CSF3, showed directionally inconsistent associations, likely reflecting differences between circulating and tissue-specific biology or compensatory responses. Overall, these findings reinforce the causal relevance of key network regulators and their targets, highlighting potential avenues for therapeutic intervention. Indeed, among 25 network regulators, 13 are druggable, with 6 already targeted by licensed or investigational drugs. Accordingly, the CPN reveals interconnected therapeutic pathways, including complement-mediated inflammation, cellular stress response, hemostatic balance, and lipid metabolism, supporting both single- and multi-pathway targeting. This framework provides a roadmap for next-generation ACVD prevention, where concordant evidence reinforces target confidence and discrepancies highlight opportunities for novel therapeutic discovery.

Among complex traits, ACVD and its associated clinical complications, including MI, stroke, and HF, represent the highest level of complexity and continue to be a leading cause of morbidity and mortality in industrialized nations. ACVD arises from the complex interplay of cardiometabolic disorders, which collectively drive the progression of coronary plaques over decades. These cardiometabolic factors, affecting the arterial wall, involve multiple tissues and are shaped by intricate genetic and environmental influences. While this high-level view of the disease has been recognized for years, the detailed endocrine signaling linking the tissues and biological processes involved remains poorly understood. The extensive profiling of 6586 unique proteins with 7523 highly sensitive aptamers has enabled exploration of the serum proteome's architecture and the regulatory relationships among blood proteins, many of which were associated with ACVD, with some demonstrating causal links to the disease. These regulatory relationships were deeply interconnected, forming cascades of protein nodes or networks that elucidate the directionality and collective mechanisms driving ACVD. Moreover, the serum protein networks span tissue boundaries, linking key tissues and organs involved in the etiology of ACVD. This work begins to reveal, at the molecular level, the cross-tissue coordination or systemic homeostasis required for disease manifestation. The initial characterization of this network lays the foundation for formulating hypotheses and directing future research on the etiology of ACVD.

## Methods

### Study population
Cohort participants aged 66 through 96 years at the time of blood collection were from the AGES[28], a single-center, prospective, population-based study of older adults ($N = 5764$, mean age $76.6 \pm 6$ years). AGES was formed between 2002 and 2006, and its participants were randomly selected from the surviving members of the established 40-year-long population-based Reykjavik study[60,61]. The Reykjavik study, a prospective cardiovascular survey, recruited a random

sample of 30,795 adults born between 1907 and 1935 who lived in the greater Reykjavik area in 1967, that were examined in six phases from 1967 to 1996[60,61]. The AGES measurements, which include for instance, brain and vascular imaging, are designed to assess four biologic systems: vascular, neurocognitive (including sensory), musculoskeletal, and body composition/metabolism[28]. All AGES participants are of European ancestry. AGES was approved by the National Bioethics Committee in Iceland, that acts as the institutional review board for the Icelandic Heart Association (approval number VSN-00-063, in accordance with the Helsinki Declaration) and by the US National Institutes of Health, National Institute on Aging Intramural Institutional Review Board.

The study comprised MI patients who met the MONICA criteria for definite MI as previously described[60]. The criteria for HF were based on clinical symptoms and signs, chest X-rays, and, in many cases, echocardiographic findings from hospital records, which were adjudicated by examining every record for both prevalent HF, i.e., had HF at the baseline visit, and incident HF, i.e., HF diagnosed after the baseline visit. The incident HF cases were free of HF diagnosis at the baseline visit, but who were later hospitalized and diagnosed (hospital discharge ICD-10 diagnosis codes starting with I50) with HF during the follow-up period of eight years. Each patient's thorough medical records were subsequently adjudicated by a cardiologist to confirm the diagnosis of symptomatic HF, and the date of the incident HF event documented. Among the criteria were symptoms such as shortness of breath that could be ambulatory, and signs of pulmonary edema. Type two diabetes (T2D) was determined from self-reported diabetes, diabetes medication use, or fasting plasma glucose ≥ 7 mmol/L according to American Diabetes Association guidelines[62]. Metabolic syndrome (MetS) was defined by three or more of the following: Fasting glucose ≥ 5.6 mmol/L, blood pressure ≥ 140/90, triglycerides ≥ 1.7 mmol/L, high-density lipoprotein (HDL) cholesterol < 0.9 mmol/L for males or < 1.0 mmol/L for females, body mass index (BMI) > 30 kg/m². Systolic and diastolic blood pressure were measured twice with subjects in a supine position using a Mercury sphygmomanometer. Lipoproteins and plasma glucose levels were measured on fasting blood samples. Triglycerides (TG) were measured using enzymatic colorimetry (Roche Triglyceride Assay Kit), HDL cholesterol with an enzymatic in vitro assay (Roche Direct HDL Cholesterol Assay Kit), and glucose was measured using photometry (Roche Hitachi 717 Photometric Analysis System). Coronary artery calcium (CAC) was quantified using the Agatston scoring method[63], which was reviewed independently by four image analysts. Phantom-adjusted CAC was expressed as a sum score for all four coronary arteries, as previously described in greater detail[64]. The use of ultrasound imaging to assess the presence and severity of carotid plaque in the AGES population has been detailed elsewhere[65,66]. Hepatic steatosis was assessed by computed tomography (CT), serving as a non-invasive proxy for non-alcoholic fatty liver disease (NAFLD), as previously described[67]. We assessed overall survival (2982 events), as well as survival following incident CHD (692 events) and incident MI (299 events). For overall survival, follow-up was defined as the time from study entry in AGES until death from any cause or the end of follow-up (December 2016). For survival after incident CHD or MI, follow-up was defined as the period starting 28 days after the incident event until death from any cause or the end of follow-up.

## Proteomics profiling

Blood samples were collected at the AGES-Reykjavik baseline visit after an overnight fast, and serum samples prepared using a standardized protocol and stored in 0.5 mL aliquots at −80 °C. Serum samples collected from the inception period of AGES, i.e., from 2002 to 2006, were used to generate proteomics data used in this study. Before the protein measurements were performed, all serum samples from this period went through their first freeze-thaw cycle. Protein levels in serum from 5376 individuals of the AGES were determined using a multiplex SOMAscan proteomic profiling platform, which employs aptamers or Slow-Off rate Modified Aptamers (SOMAmers) that bind to target proteins with high affinity and specificity. Here, 7523 aptamers mapping to 6,586 UniProt IDs were measured in a total of 8592 samples (two time points) using the SomaScan_v4.1 platform[68]. The aptamer-based platform measures proteins with femtomole (fM) detection limits and a broad detection range ( > 8-log dynamic range) of concentration[69]. To prevent biases related to batch or processing time, the order of sample collection and separate sample processing for protein measurements were randomized, and all samples run as a single set at SomaLogic Inc. (Boulder, CO, US). All aptamers that passed quality control had median intra-assay and inter-assay coefficient of variation, CV < 5%. Hybridization controls were used to correct systematic variability in detection and calibrator samples of three dilution sets (20% (1:5), 0.5% (1:200), and 0.005% (1:20,000)) were included so that the degree of fluorescence was a quantitative reflection of protein concentration. The adaptive normalization by maximum likelihood (ANML) method was employed to normalize Quality Control (QC) replicates and samples using point and variance estimations from a healthy external reference population ($n = 1000$). Consistent target specificity of aptamers was indicated by direct (through mass spectrometry) and/or indirect validation[23].

Some proteins were targeted by more than one aptamer. In such cases, individual aptamers had distinct binding sites (epitopes) or binding affinity[23]. The single gene *NPPB*, for example, produces three protein products: full-length BNP, NT-proBNP, and BNP32, each of which are targeted by different aptamers. Duplicate aptamers to single pass transmembrane proteins (one to the extracellular domain and another to the intracellular loop), aptamers targeting multimers (e.g., interleukins), and duplicate aptamers generated in distinct expression systems are further examples. Finally, 233 aptamers were derived from mouse-human protein chimeras (used as SELEX input) to target proteins from both species.

Before the analyses, protein data were centered, scaled, and Box-Cox transformed[70], and extreme outlier values were excluded, defined as values above the 99.5th percentile of the distribution of 99th percentile cutoffs across all proteins. Prior to reconstruction of the causal protein networks, the data were adjusted using a linear model to account for age and sex.

## Genotype data and the detection of cis-acting variants

The genotype data includes assayed and imputed genotype data for 5656 AGES participants[71]. The genotyping arrays used were Illumina Hu370CNV and Illumina GSA BeadChip, which were quality controlled by eliminating variants with call rates < 95% and Hardy Weinberg Equilibrium (HWE) $P$-value $< 1 \times 10^{-6}$. The arrays were imputed against the Haplotype Reference Consortium imputation panel r1.1, and post-imputation quality control was performed separately for each platform. Variants with imputation quality $r^2 < 0.7$, minor allele frequency < 0.01, as well as monomorphic and multiallelic variants, were removed before merging the platforms to generate a dataset with 7,506,463 variants for 5656 AGES individuals as previously described[71]. These variants were associated to each of the aptamers on the v4.1-7k serum protein panel to identify proximal (*cis*-acting) pQTLs, in the same way as previously described[71]. We applied a 300 kb genomic window spanning each protein-expressing gene in the v4.1-7k serum protein panel to map out *cis*-acting pQTLs after accounting for the number of single-nucleotide polymorphisms (SNPs) in each window. We then corrected for multiple testing using the Storey-Tibshirani procedure for q-value estimation[72]. The *cis*-acting pQTLs serve as instrumental variables for the reconstruction of the causal serum protein networks, which are described below.

## Reconstruction of the circulating causal protein network

A circulating Causal Protein Network (CPN) was reconstructed from causal relationships between serum proteins derived using a Mendelian Randomization (MR) framework[73]. In this model, causal relationships between protein pairs are estimated between an exposure protein (*A*) and outcome protein (*B*), using a *cis*-acting pQTL (*E*) for *A* as a causal instrument. We selected all proteins which had at least one valid *cis*-acting pQTL as our A-proteins at FDR ≤ 5%, after correcting for multiple testing using the Storey-Tibshirani procedure[72]. In each pairwise causal test, the lead protein regulatory SNP (pSNP) for *A* (lowest *P*-value) was selected as *E* to be used as an instrumental variable in accordance with the core instrumental variable assumptions[73,74]: (1) the instrumental variable should be strongly associated with the exposure (*A*), (2) the instrumental variable should only be associated with the outcome (*B*) through the exposure (*A*), and (3) the instrumental variable should not be associated with any potential confounders affecting the exposure (A) and the outcome (B).

Causal estimates between proteins were measured using the tool Findr[20] (version 1.0.8) in Python (version 3.10.6) using individual-level protein expression levels for *A* and *B* and genotypes for *E*. Before inferring casual estimates, the data were transformed using a rank-based inverse normal transformation within the Findr package. The product of the secondary linkage test (P2) and controlled test (P5) from Findr[20] was used to estimate the posterior probability (PP) of P($A \to B$). The secondary linkage test measures the PP of association between *E* and B, and the controlled test assesses the PP of the dependence between *A* and *B* following adjustment with E to exclude that E has independent effects on A and B[75]. We estimated P($A \to B$) for all A-proteins with a valid instrumental variable, where *B* was every other protein in our dataset. We estimated a global FDR as 1 minus the mean of all PP for P($A \to B$) and then filtered PP P($A \to B$) to achieve the desired FDR threshold, as shown previously[76]. Networks were reconstructed from causal interactions that fell below this FDR threshold, where parent nodes were *A* and child nodes were *B*, with edges represented as P($A \to B$). In instances where there were multiple aptamers targeting the same protein, we selected the *A* proteins with the largest number of targets as the representative aptamer for this protein. When examining hierarchy between regulators, we converted edges between *A* proteins to a directed acyclic graph (DAG) using a greedy heuristic as implemented in Findr[20], which selects the most significant edges in an iterative fashion to avoid cycles. This is done to identify any hierarchical structure between A-proteins, and once a DAG has been constructed the removed edges can be reintroduced to examine how well any hierarchical structure is preserved within the complete networks. Network visualization was performed using Cytoscape (version 3.10.1).

As pleiotropy is a concern in MR analyses, we took steps to account for instances where two or more exposure proteins either shared or had instrumental variables in high linkage disequilibrium (LD)[74]. We assigned proteins to the same LD block if they shared the same lead pSNP, or if they had different pSNPs in medium or high LD ($r^2 \geq 0.5$). For all *A* proteins within an LD block, we calculated the intersection of targets as a ratio of the union of targets (*I*) in a pairwise manner between target sets. We resolved *A* proteins as independent networks when *I* < 0.6, in cases where *I* > 0.6 we collapsed the union of all targets within a single unresolved network. In cases where two or more *A* proteins were mutual targets of each other, we considered these networks as unresolved. Where there were more than two A-proteins in a single LD block, an A-protein must have I < 0.6 across all other A-proteins to be considered independent. We did not encounter any instances of A-proteins that were shared between separate LD blocks.

We defined protein subnetworks as a single A-protein and all targets of that A-protein, i.e., a single regulator and its targets. The average expression profile of a subnetwork was estimated as an eigenvector of all proteins in a subnetwork across all samples, described here as an eigenprotein[77]. We calculated the first principal component (PC) from the expression profiles of all proteins within a subnetwork using principal component analysis (PCA), via the PCA function from the Python library scikit-learn[78] (version 1.1.2). If PC1 explained >15% of the variance of the subnetwork, we used this principal component as an eigenprotein for this subnetwork in downstream analyses, and for subnetworks where the variance was <15%, PC1 was not taken forward. Three CPN subnetworks did not meet this criterium and therefore did not have a valid eigenprotein.

## Network robustness analyses

To examine the impact of varying sample size on network inference, we randomly sampled subsets of individuals in different sizes for the reconstruction of networks. At each sample size threshold ($n = 500$, 1000, 2000, 3000, 4000, 5000), we randomly sampled three subsets from the pre-processed protein expression data and reconstructed a network for each subset using the previously outlined approach. We termed the primary network, which was reconstructed using all samples, as the ground-truth network. The ground-truth network was represented as a flattened binary matrix (A,B), where 1 was indicative of the presence of an edge between A → B, and 0 for the absence of an edge. Each sub-sampled network was represented as a similar flattened matrix (A,B), but where the values were populated by the PP of P(A → B). We tested how well each sub-sampled network captured the structure of the ground-truth network by calculating the Receiver Operating Characteristic Area Under the Curve (ROC AUC) for each sub-sampled network against the ground-truth network using the roc_auc_score function from scikit-learn (version 1.1.2)[78]. We then calculated the mean AUC of the repeated sub-sampled networks at each sample size threshold. We also calculated Precision-Recall for each sub-sampled network against the ground-truth network using the precision_recall_curve function, also from scikit-learn[78].

## Variance explained model

To identify independent *cis*-acting protein SNPs, a ± 150 kb window was specified around each lead variant to define the region of interest for association testing. Using the GCTA software (version v1.94.1)[79], the cojo-slct parameter was applied using a forward model selection approach. The default collinearity cutoff of 0.9 was used, and the *P*-value threshold was set at 0.00763, which corresponds to the highest *P*-value maintaining an FDR below 5%. Having identified independent *cis*-acting protein SNPs for every network regulator, we estimated the proportion of variance in protein expression that could be explained by *cis*-acting pQTLs using multiple linear regression. For each protein, we fitted a linear model in R (version 4.3.2), where the genotypes for the independent *cis*-acting SNPs act as the explanatory variables for protein expression. The adjusted coefficient of determination (adjusted $r^2$) from this model was used as an estimate of the variance explained by the *cis* component for each protein, and the mean adjusted $r^2$ was then calculated across all network regulators.

We also estimated the variance in target protein expression that could be explained by *cis*-acting pQTLs from the target regulators. More specifically, the influence of *cis* signals on network regulators (parental nodes) was assessed in relation to the expression of 5459 target proteins within the CPN, including 162 target proteins that also served as regulators for other proteins. For every target protein, we fitted a linear model, as described previously, using the genotypes of all independent *cis*-acting pQTLs for the regulators (parental nodes), in addition to any *cis*-acting pQTLs for the target itself. We then calculated the difference in variance explained by local and parental *cis*-acting pQTLs combined, to that explained by *cis*-acting genetic

variation alone. If the target protein did not have any *cis*-acting pQTLs, then the variance explained by the *cis* component was set to 0. Any unresolved networks were excluded from this analysis.

## Comparison with a reference database of protein-protein interactions

We used the human integrated protein-protein interaction reference database (HIPPIE) version 2.3[33] to identify experimentally derived protein-protein interactions (PPIs) that have been captured by the serum CPN described in this study. We accessed all 289,112 PPIs from the HIPPIE database, which have been scored as a weighted sum, based on the reliability of the evidence underpinning the interactions and the number of studies detecting each interaction. There are 262,346 PPIs scored at medium confidence (confidence score > 0.63) and 77,630 scored at high confidence (confidence score > 0.72), based on thresholds defined by HIPPIE authors. We then calculated the overlap between edges in the CPN with PPIs from HIPPIE at the different confidence thresholds. After which, we compared the number of common interactions between HIPPIE and the CPN to common interactions between HIPPIE and the edges from random networks. These networks were generated by randomly sampling proteins from the complete set of measured AGES proteins to produce the same number of edges as in the CPN. This process was performed 10 times, and the mean number of edges captured, in addition to the standard deviation, was calculated at each of the different confidence thresholds. We then calculated a z-score comparing the number of edges captured from the true vs random networks at each confidence threshold as: z = (observed overlap - mean random overlap) / stdev random overlap. This was then converted to a *P*-value as: *P*-value = 1 – cdf(z-score), using the norm.cdf() function from Scipy Stats[80].

## Network topology analyses

We identified weakly connected components (i.e., groups of nodes in the CPN where every node can be reached from any other node, regardless of the direction of the edges) in the complete CPN and after removal of the regulators with the most targets, applying the weakly_connected_components function from the Graphs.jl (v1.12.0) package using Julia v1.11.1. We obtained a hierarchical layout of the A-protein DAG by first defining root nodes as A-proteins without incoming edges. We then defined the level of each A-protein as the shortest distance (number of edges in a shortest path) from a root node to the protein of interest. Shortest paths were computed using the dijkstra_shortest_path function from the Graphs.jl (v1.12.0) package using Julia v1.11.1. To further simplify the structure of the DAG, we performed transitive reduction using the "transitive reduction" function from the Graphs.jl (v1.12.0) package from Julia. Transitive reduction reduces the original DAG G to a new DAG G' with the fewest possible edges such that, if there is a path from a vertex x to a vertex y in G, there must also be a path from x to y in G', and vice versa.

## Validating the network structure across proteomic platforms and cohorts

The UKBB is a prospective population study of 502,128 individuals from the UK who have been extensively characterized by genomic and phenotypic traits[39]. Genotype data is available for 486,593 participants, obtained using either the Applied Biosystems UK BiLEVE Axiom Array by Affymetrix or the Applied Biosystems UKBB Axiom Array. Imputation was carried out using the HRC reference panel, yielding allelic dosages for ~96 million variants, with markers annotated using the GRCh37 assembly of the human genome[58].

There are 53,013 (53.9% female, mean age = 56.8 years) UKBB participants who have undergone plasma protein profiling for 2923 proteins using the antibody affinity based Olink Explore 3072 protein extension assay platform[59]. We processed the Olink assay data using

the approach outlined by Sun et al.[59], from the initial study reporting plasma measurements from UKBB, whereby assays with more than 20% missingness are removed and the remaining missing values are mean imputed across samples. The overlap between processed samples and those with corresponding genotype data yielded 52,543 samples to be taken forward for network reconstruction. The remaining protein assays were then filtered for those that had a corresponding assay in the AGES SOMAscan 7K panel, yielding 2186 comparative protein assays. These assay measurements were then corrected for age and sex by fitting to a linear model as previously described for the AGES proteomic data.

Due to differences between the SOMAscan and Olink platforms, instrument selection was carried out independently for the UKBB proteomic data. For each protein, the lead *cis*-acting pSNP was chosen based on the *P*-value, using pQTL summary statistics from Sun et al.[59]. Reconstruction of causal protein networks was carried out with Findr and edges selected based on FDR thresholds, as has been previously described for the AGES discovery cohort. We tested the significance of overlapping targets between comparative subnetworks by hypergeometric testing using the hypergeom() function from Scipy Stats[80] in Python, while using the intersection of measured proteins between platforms as a background. We then corrected for multiple testing using the Storey-Tibshirani procedure for q-value estimation[72].

## Colocalization and Mendelian randomization analysis

We used colocalization analysis to identify *cis*-acting pQTLs that shared a common signal with six different phenotypic traits using the R package coloc (version 5.2.3)[81] and publicly available GWAS summary statistics from individuals with European ancestry (Supplementary Data 10). We first identified all proteins that had a *cis*-acting pQTL (FDR < 5%) that shared at least 1 common *cis*-SNP with the GWAS trait (*P* < 5 × 10⁻⁸) and extracted the summary statistics for all SNPs within ±150 Kb of the transcription start site for the cognate encoding gene. We then estimated the probability of a shared causal variant (PP.H4) using approximate Bayes Factor colocalization[82] via the coloc.abf (coloc) function using a threshold of PP > 0.9 and > 0.5. In addition, we tested colocalization between overlapping signals of cardiometabolic GWAS traits and serum protein *cis*-acting pQTL using the Bayes Factor colocalization with the Sum of Single Effects (SuSiE) framework for fine mapping of genomic loci[83]. The SNPs within the *cis* window were fine-mapped for both the GWAS and pQTL signal using the Susie rss() function from the SusieR R package (version 0.12.35), while limiting the number of possible independent signals (*L* = 5) and the maximum number of iterations (*n* = 5000). Only loci where convergence was observed for both signals were taken forward for colocalization. Colocalization was performed using the coloc.susie() function from the Coloc R package (version 5.2.3), which performs Bayes Factor colocalization. We selected colocalized signals (H4) using the posterior probability thresholds > 0.9 and > 0.5.

The causality of selected network regulators was assessed using a forward two-sample MR approach[84]. Genetic instruments (*cis*-acting pSNPs) were identified within ±150 kb of the protein-coding region based on AGES data, filtered for statistical significance (*P* < 0.05/ number of *cis*-region SNPs), and matched to external GWAS outcomes. These SNPs were clumped for LD (*r*² < 0.1 and < 0.2) using PLINK v1.9[85], and AGES genotype data. MR estimates were calculated using the Wald ratio for single-SNP associations and generalized weighted least squares (GWLS) for multi-SNP analyses as previously described[86]. Significant results (FDR < 0.05) were deemed causal candidates. To ensure the robustness of our findings, we performed sensitivity analyses of the MR estimates using MR-Egger[87] and weighted median estimation[88]. For proteins with more than three instruments available, instrument heterogeneity was assessed using Cochrans´s Q[89]. Horizontal pleiotropy was assessed using MR-Egger[87]. The generalized

GWLS was performed as previously described[90], while other MR analyses were performed using the TwoSampleMR[91] R package.

## Statistical analysis and functional enrichment analysis

The relationship between serum protein (and eigenprotein) levels and quantitative phenotypes was evaluated using linear regression controlling for age and sex, whilst the relationship between serum protein (and eigenprotein) and prevalent disease was examined cross-sectionally using logistic regression with age and sex adjustments. The associations between serum proteins (and eigenprotein) and incident disease were evaluated longitudinally using the Cox proportional-hazards model[92], with a median follow-up period of 5.6 [2.8, 8.2] years for incident MI. Associations with Benjamini-Hochberg FDR < 0.05 were considered statistically significant. To identify enriched gene ontology (GO) terms and pathways within network targets, we carried out formal function enrichment analysis using gprofiler2 (version 0.2.2), the R interface of g:Profiler (R version 4.3.2). We used a custom background of all measured proteins in the AGES aptamer-based assay and corrected for multiple testing using Benjamini-Hochberg correction with a cut-off of FDR < 0.05 to identify any enriched terms across the following categories; GO:MF, GO:BP, GO:CC, KEGG, REACTOME, Wikipathways, miRTarBase, Human Protein Atlas and Human Phenotype Ontology.

## Reporting summary

Further information on research design is available in the Nature Portfolio Reporting Summary linked to this article.

## Data availability

Data from the AGES Reykjavik study are available through collaboration (AGES_data_request@hjarta.is) under a data usage agreement with the IHA. All access to data is controlled via the use of subject-signed informed consent authorization. The time it takes to respond to requests varies depending on their nature and the circumstances of the request, but it will not exceed 14 working days. All data supporting the conclusions of the paper are presented in the main text and freely available through supplementary data to this manuscript. Access to the UK Biobank was obtained through project ID 102820. An MTA was signed with the UK Biobank, agreeing to comply with all terms of use. The UK Biobank data are available upon application (www.ukbiobank.ac.uk).

## Code availability

All code used in this study is accessible at the following repository: https://github.com/sbankier/AGES_causal_protein_networks with the[93] under an MIT license.

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

## Acknowledgements

The authors acknowledge the contribution of the Icelandic Heart Association (IHA) staff to the AGES-RS, as well as the involvement of all study participants. This research has been conducted using the UK Biobank Resource under Application Number 102820. The proteomics work was carried out in collaboration with Novartis Biomedical Research (NIBR). National Institute on Aging (NIA) contracts N01-AG-12100 and HHSN271201200022C for V.G. financed the study. V.G. received funding from the NIA (1R01AG065596-01A1), and IHA received a grant from the Icelandic Parliament. T.M. acknowledges support from the Research Council of Norway (project number 312045), the European Union's Horizon Europe (European Innovation Council) program (grant agreement number 101115381), and the L. Meltzers Høyskolefond.

## Author contributions

S.B and V.E co-wrote the manuscript. S.B., V.E., and T.M. produced visualizations and contributed to the conception and design of this research. S.B., V.E., T.M., Va.G., H.B., and T.J. performed the analyses. S.B., Va.G., and T.J. were responsible for data curation. V.E., T.M., and V.G. co-supervised the work. All other authors, including J.L., L.W., E.A.F., N.F., L.J.L., J.L.M.B., L.L.J., T.A., A.P.O., and J.R.L., contributed to data interpretation, manuscript review, and editing. All coauthors have approved the submitted version of the paper.

## Funding

## Competing interests

J.L., L.L.J., N.F., and A.P.O. are employees and stockholders of Novartis. All other authors have nothing to disclose.
