## [Transparent Peer Review file · Nature Communications]

Circulating causal protein networks linked to future risk of myocardial infarction

Corresponding Author: Professor Valur Emilsson

Version 0:

Reviewer comments:

Reviewer #1

(Remarks to the Author)

In this manuscript, Bankier, Gudmundsdottir and colleagues present an impressive large-scale proteomic study of 5,376 older adults from the AGES-Reykjavik cohort, in which 7,523 serum proteins were quantified. The authors reconstruct a global causal network of circulating proteins and examine their relationships with cardiometabolic traits, including myocardial infarction (MI) and heart failure (HF).

The manuscript is clearly written and presents a substantial contribution to the field. The findings are likely to inform efforts in drug development and repurposing for cardiometabolic disease. That said, I believe the study could be further strengthened through several methodological refinements and additional contextual discussion, as outlined below.

Major comments

- 1) Replication. The protein network reconstruction is data-driven and could be sensitive to cohort-specific features. Has the team considered validation in an independent cohort? Confirmation of the network structure or key associations in an external dataset would considerably enhance the robustness and generalisability of the findings.
- 2) One-sample Mendelian Randomisation (MR) and risk of bias The use of one-sample MR introduces potential for bias and inflation of the type I error rate (see: Burgess et al., Wellcome Open Res, 2019;4:186). Where feasible, a two-sample MR approach would be preferable, drawing upon publicly available GWAS summary statistics for the cardiometabolic traits in question. This would reduce bias, increase statistical power, and facilitate a range of sensitivity analyses (e.g., MR Egger, weighted median) that are vital for assessing the robustness of causal inference.
- 3) Colocalization methodology. The authors may wish to consider the application of more advanced colocalization approaches capable of modelling multiple causal variants, such as SuSiE or PwCoCo (e.g. PMID: 35452592). These methods are better suited to complex loci and may yield more reliable insights regarding shared genetic aetiology between proteins and traits.
- 4) Inclusion of positive controls. It may be instructive to include a small number of well-established biological pathways or protein complexes—such as those involved in transcriptional regulation, the cell cycle, or intracellular transport—as positive controls. Demonstrating that the network approach successfully identifies such relationships would lend confidence to the method's validity.
- 5) Ancestry and generalisability As the study sample is limited to individuals of European descent, the authors should more clearly acknowledge this as a limitation. While expansion to more diverse populations may not yet be feasible, explicit recognition of this issue will help orient future work towards greater inclusivity and global relevance.

Minor comment

Line 540: It would be helpful to define the symbol “E” for the benefit of readers less familiar with network analysis terminology.

In sum, this is a high-quality manuscript addressing an important topic, with clear translational relevance. The above suggestions are intended to strengthen the rigour and transparency of the work. I look forward to seeing the revised version.

(Remarks on code availability)

Reviewer #2

(Remarks to the Author)

The manuscript presents a sophisticated causal network inference framework leveraging 7,523 serum proteins measured in 5,376 elderly individuals from the AGES-Reykjavik Study. The rationale for a system-level approach and the use of pQTLs for causal protein network inference is well-articulated, and the methodology is technically sound but highly complex. Below are specific concerns and suggestions for improvement.

1. Abstract

The statement "we used cis-acting serum protein quantitative trait loci (pQTLs) as instrumental variables, while accounting for potential unobserved confounding factors" requires clarification. If this refers to Mendelian randomization (MR), the phrasing is misleading—MR mitigates unobserved confounding via genetic IVs but does not explicitly "account" for it. Please revise to reflect MR's assumptions accurately.

2. Results (Line 129–130)

"The follow-up period for newly diagnosed MI patients extended up to 12 years from baseline, with recurrent cases excluded from the incident group analysis. Person-years of follow-up were calculated from the first AGES visit until the date of diagnosis, death, or the end of the follow-up period..." Incident vs. Recurrent Cases: If follow-up was censored at the first MI diagnosis (i.e., incident cases only), recurrent MI cases should not exist in this cohort by definition.

3. Discussion

The clinical implications of prioritized proteins (Table 2) are underdeveloped. Given the focus on ASCVD early detection and prevention, readers would benefit from:

- A brief summary of how many proteins have licensed/investigational drugs targeting them.
- Potential therapeutic pathways suggested by the causal network.

4. Methods

Clarify definition of NAFLD/MASLD criteria.

Genomic Window Inconsistencies: Varying window sizes are used across analyses (300 kb for cis-pQTL detection, 150 kb for variance modeling/colocalization, 500 kb for MR IVs). Justify these choices.

Colocalization Assumptions: The analysis assumes a single causal variant. For robustness, consider methods accommodating multiple causal variants (e.g., PWCoCo, SharePro) for prioritized proteins.

The LD clumping threshold ($r^2 \geq 0.2$) is lenient and may retain non-independent cis-pQTLs. Address whether stricter thresholds (e.g., $r^2 < 0.1$) is not feasible here.

(Remarks on code availability)

Reviewer #3

(Remarks to the Author)

Summary

This manuscript by Emilsson and colleagues presents a comprehensive systems biology approach to understand the molecular outcomes underlying myocardial infarction (MI) and associated cardiometabolic conditions. By analyzing serum proteomic data through Somascan with samples from the AGES-Reykjavik study and employing a causal inference framework using cis-acting pQTLs, the authors reconstruct 185 high-confidence causal protein subnetworks (CPNs). These subnetworks were then associated with incident MI, metabolic syndrome, heart failure, and related traits. The authors attempted to validate their network using internal metrics and reference protein-protein interaction databases, and lastly performed colocalization and Mendelian Randomization (MR) analyses in order to strengthen their claims.

I think that this study represents an important advancement in causal proteomics and has the potential to uncover novel biomarkers and therapeutic targets for cardiovascular disease. The greater than 5,000 samples afforded by the AGES cohort is compelling. However, some conceptual and methodological weaknesses should be addressed to enhance the study's impact and interpretability. Importantly, the analytical framework is strong, but the study would benefit from independent validation, greater biological resolution, and experimental follow-up.

Major Strengths

The AGES-Reykjavik cohort provides deep phenotyping and proteomic data, with sufficient statistical power to explore complex biological relationships because of this large, richly annotated patient cohort. This is considered a high profile strength of the project

The application of Mendelian Randomization and the Findr pipeline to reconstruct directed protein networks from high-dimensional serum data is well-justified and methodologically sound. And represents an innovative use of causal inference.

The authors conduct extensive robustness tests, including AUC/precision-recall analysis, transitive reduction, and overlap with known PPIs, lending feasibility and credibility to their findings.

Associations with future MI and HF events, alongside known clinical traits, support the clinical applicability of the identified network regulators (e.g., ITIH3, APOA5) showing translational potential

Major Weaknesses and Areas for Improvement

Lack of External Validation. The causal protein networks are derived from a single cohort of elderly Icelandic participants. No independent replication is attempted in other populations or platforms. It is entirely possible to validate key subnetworks or associations (e.g., ITIH3, DDX39B, APOA5) in an external proteomic cohort even if limited in number, or more relevant to a multi-ethnic cohort. It is entirely possible as well, that there are public available datasets, such as the UK biobank which could serve as a validation informatic approach.

Informatic analysis/Indirect Edge Ambiguity and Confounding interpretation. While transitive reduction is applied to remove some indirect edges, a large number of network edges may still reflect indirect regulation or residual confounding. A recommendation might be to incorporate formal mediation models or latent confounder correction to further refine causal edge specificity.

Functional Validation Missing.

The study is computational in nature; no experimental follow-up or perturbation assays are included to confirm network-derived predictions. Here, a suggestion would be to target a subset of high-priority regulators (e.g., ITIH3, HSPA1A, LRP4) for validation using in vitro or in vivo models, or at minimum, review existing functional studies in greater detail.

There is some potential ambiguity in protein-aptamer mapping, particular due to the platform that the authors used. Somascan has some inherent technical limitations. In this study, several aptamers map to the same protein family, sometimes targeting different isoforms or protein domains. Such dual identification could result in artificial interpretation or duplication of network regulators. It is recommended that authors clarify how redundancy in aptamer targeting was handled, and consider collapsing overlapping aptamers to reduce spurious edges.

Phenotypic Associations and Effect Size Interpretation. While statistically significant, the magnitude of associations between network eigenproteins and traits is not always clinically contextualized. The authors are suggested to maybe report hazard ratios or beta coefficients with interpretation to support translational relevance.

Minor Point

The study is missing "Negative Controls". For instance, it would strengthen the interpretation to report null associations (e.g., subnetworks NOT linked to MI or metabolic traits) to contextualize specificity, this is important context to the manuscript.

(Remarks on code availability)

Version 1:

Reviewer comments:

Reviewer #1

(Remarks to the Author)

The revised manuscript is much stronger, particularly due to the replication in UKBB and the SuSiE colocalization analysis. I have no objections to its acceptance for publication in Nature Communications.

(Remarks on code availability)

N/A

Reviewer #2

(Remarks to the Author)

None.

(Remarks on code availability)

Reviewer #3

(Remarks to the Author)

I have no additional concerns

(Remarks on code availability)

Response to Review

We are pleased to submit the revised version of our manuscript, entitled “*Circulating causal protein networks linked to future risk of myocardial infarction*” (NCOMMS-25-18281-A), for consideration for publication in *Nature Communications*. We sincerely thank the reviewers for their thoughtful and constructive feedback. In the accompanying point-by-point response document, we have carefully addressed each of the comments. These revisions also include correction of typographical errors, and adjustments to ensure full compliance with *Nature Communications* formatting guidelines. We believe that these changes have substantially strengthened the manuscript.

Reviewer comments are addressed in blue font in the response document. Additions to the revised manuscript are indicated in *italicized text*. Line numbers cited in our responses refer to the clean version of the revised manuscript, submitted alongside a tracked-changes version highlighting all edits.

Reviewer comments

Reviewer #1:

In this manuscript, Bankier, Gudmundsdottir and colleagues present an impressive large-scale proteomic study of 5,376 older adults from the AGES-Reykjavik cohort, in which 7,523 serum proteins were quantified. The authors reconstruct a global causal network of circulating proteins and examine their relationships with cardiometabolic traits, including myocardial infarction (MI) and heart failure (HF).

The manuscript is clearly written and presents a substantial contribution to the field. The findings are likely to inform efforts in drug development and repurposing for cardiometabolic disease. That said, I believe the study could be further strengthened through several methodological refinements and additional contextual discussion, as outlined below.

We thank the reviewer for his/her positive remarks.

Major comments

1) Replication. The protein network reconstruction is data-driven and could be sensitive to cohort-specific features. Has the team considered validation in an independent cohort? Confirmation of the network structure or key associations in an external dataset would considerably enhance the robustness and generalisability of the findings.

Response: While we previously acknowledged the current scarcity of external datasets suitable for our network structure replication, we fully agree with the Reviewer that independent validation would strengthen the conclusion of our study. Reviewer 3 mirrors Reviewer 1’s concern (see below) and specifically recommends testing replication using the UK Biobank proteomics data. We therefore evaluated the causal protein network architecture in the UK Biobank Olink dataset, comprising 52,543 samples and 2,186 proteins that overlap with those measured in the AGES study, despite the previously noted modest concordance with our SomaLogic aptamer-based measurements (including only moderate Spearman correlations and discrepancies in *cis*-acting pQTL signals for many analytes)¹. Accordingly, while performing this analysis, we have made a point of emphasizing platform heterogeneity as an important caveat for interpreting cross-cohort

comparability. Based on this analysis, we have added new Supplementary Table S10 and Supplementary Fig. S11 (displayed at the end of this document), along with corresponding text in the Results, Discussion, and Methods sections as detailed below:

Lines 297-337 in the Result section:

“Replication of causal protein network architecture in an independent cohort

“We performed a validation analysis of the CPN structure using the UK Biobank (UKBB) study² as an independent dataset. There are 2,186 assays from the UKBB Olink Explore and the AGES 7K SOMAscan profiling platforms that target the same proteins according to their annotations (Methods). Therefore, we reconstructed CPNs in both AGES and UKBB using only this subset of proteins from each platform to ensure that these networks were comparable. As correlations in protein measurements between these two different proteomic platforms have been shown to vary³, we conducted instrument selection separately in the UKBB and AGES studies (Methods), removing duplicate aptamers targeting the same protein before analysis.

We first reconstructed a subsetting CPN in AGES but restricted it to the 2,186 proteins also measured in UKBB. Of these protein assays, 1,835 had a valid cis-acting pQTL instrument to be used as a potential network regulator. At a 1% FDR threshold we identified 454 proteins with at least one target and 43 with at least 10 targets in the AGES study. Among the 454 proteins with at least one target in the subsetting AGES CPN (FDR = 1%), 416 were also present in the full AGES CPN constructed from all measured proteins. Furthermore, 96.3% of the edges identified in the subsetting AGES CPN overlapped with those in the full CPN. We then reconstructed a comparative CPN using data from 52,543 UKBB participants with Olink assays targeting the same 2,186 proteins. We observed a significant overlap between network regulators in the two cohorts, including the 454 proteins with at least one target in AGES, of which 393 were present with at least one target in UK Biobank (Fisher’s Exact Test; OR = 1.9, $P = 4.1 \times 10^{-6}$). Further, we identified 629 edges shared between the AGES subsetting and UK Biobank CPNs (FDR = 1%). To formalize this comparison, we applied Fisher’s Exact Test, which demonstrated that the overlap was highly significant (OR = 2.9, $P = 1.6 \times 10^{-99}$), using all possible edges as the background while excluding self-edges. Notably, many more targets were identified for a given CPN in UKBB than in AGES (Supplementary Table S10), likely to reflect the nearly tenfold larger sample size in UKBB.

We next sought to determine the extent to which protein targets of subnetworks reconstructed in AGES and those from UKBB overlapped. Of the 393 shared network regulators from the comparative subnetworks, we found 36 that shared at least 2 targets in the AGES and UKBB comparative CPNs, for which we assessed overlap significance with a hypergeometric test, using the full set of 2,186 proteins as background. Overall, 23 subnetworks had a significant overlap in targets across both platforms ($q < 0.05$), including three of the top-ranked subnetworks previously linked to ACVD in the full AGES CPN, namely, KLKB1, CFB and CSF3 (Supplementary Table S10, Supplementary Fig. S11).

A recent study compared correlations between Olink and SOMAlogic platforms in the Icelandic population³. From this study, we obtained correlations for 1,677 of the 2,186 proteins measured (mean Pearson coefficient = 0.39). For 19 of the 23 significantly overlapping subnetworks, correlations were available, with a higher mean of 0.48 (Supplementary Table S10). In some cases, network regulators of significantly overlapping CPNs were not correlated across platforms (Supplementary Table S10), suggesting that even when different affinity-based technologies bind

distinct epitopes, the network regulator remains robust and continues to capture the same underlying biology reflected in their downstream targets.”

Lines 478-493 in the Discussion section: *“Validating our CPNs is inherently challenging because complementary external data are scarce, unlike tissue-specific gene regulatory networks that can be supported by gene expression QTLs and external transcription factor-target datasets⁴. To address this challenge, we drew on existing resources such as the UKBB^{5,6}, although its protein coverage is considerably smaller than in our study, and inter-platform correlations are modest (mean $r = 0.39$), with additional differences arising from the epitope targeting inherent to affinity-based technologies. Nevertheless, the identification of 23 significantly overlapping subnetworks (out of 36 comparable networks) across AGES and UKBB highlights that the CPN architecture is robust. We further observed strong cross-cohort replication of global regulators and their connections, with regulators and edges significantly overlapping beyond chance expectations. The UKBB’s tenfold larger sample size facilitated the detection of more targets per regulator, but its reduced proteomic coverage limited a more comprehensive replication of networks, emphasizing the dual importance of sample size and proteomic breadth for network reconstruction and comparison. Importantly, several of the top-ranked subnetworks, including KLKB1, CFB, and CSF3, were consistently replicated across platforms and populations, underscoring their biological relevance beyond platform-specific effects.”*

Lines 774-799 in Method section:

“Validating the network structure across proteomic platforms and cohorts

The UKBB is a prospective population study of 502,128 individuals from the UK who have been extensively characterized for genomic and phenotypic traits². Genotype data is available for 486,593 participants, obtained using either the Applied Biosystems UK BiLEVE Axiom Array by Affymetrix or the Applied Biosystems UKBB Axiom Array. Imputation was carried out using the HRC reference panel, yielding allelic dosages for ~96 million variants, with markers annotated using the GRCh37 assembly of the human genome⁵.

There are 53,013 (53.9% female, mean age = 56.8) UKBB participants who have undergone plasma protein profiling for 2,923 proteins using the antibody affinity based Olink Explore 3072 protein extension assay platform⁶. We processed the Olink assay data using the approach outlined by Sun et al.⁶, from the initial study reporting plasma measurements from UKBB, whereby assays with more than 20% missingness are removed and the remaining missing values are mean imputed across samples. The overlap between processed samples and those with corresponding genotype data yielded 52,543 samples to be taken forward for network reconstruction. The remaining protein assays were then filtered for those that had a corresponding assay in the AGES SOMAscan 7K panel, yielding 2,186 comparative protein assays. These assay measurements were then corrected for age and sex by fitting to a linear model as previously described for the AGES proteomic data.

Due to differences between Somalogic and Olink platforms, instrument selection was carried out independently for the UKBB proteomic data. For each protein, the lead cis-acting pSNP was chosen based on the P-value, using pQTL summary statistics from Sun et al.⁶. Reconstruction of causal protein networks was carried out with Findr and edges selected based on FDR thresholds,

as has been previously described for the AGES discovery cohort. We tested the significance of overlapping targets between comparative subnetworks by hypergeometric testing using the hypergeom() function from Scipy Stats⁷ in Python, while using the intersection of measured proteins between platforms as a background. We then corrected for multiple testing using Storey-Tibshirani procedure for q-value estimation⁸.”

2) One-sample Mendelian Randomisation (MR) and risk of bias The use of one-sample MR introduces potential for bias and inflation of the type I error rate (see: Burgess et al., Wellcome Open Res, 2019;4:186). Where feasible, a two-sample MR approach would be preferable, drawing upon publicly available GWAS summary statistics for the cardiometabolic traits in question. This would reduce bias, increase statistical power, and facilitate a range of sensitivity analyses (e.g., MR Egger, weighted median) that are vital for assessing the robustness of causal inference.

Response: Two-sample Mendelian randomization (MR) was conducted and clearly described in both the Results and Methods sections of the current manuscript. See for instance current Result section: “We performed colocalization and two-sample MR analysis on the network regulators listed in Table 2 to further investigate potential causal relationships with ACVD related outcomes (see Methods)”, and Method section: “The causality of selected network regulators was assessed using a two-sample *cis*-Mendelian randomization (MR) approach...”

While sensitivity analysis was not included in the previous version, it has now been performed and is presented in a new Supplementary Table S14, with corresponding methodological details added to the Results and Methods section. See also Reviewer 2 comments on MR below, related to the use of different LD clumping thresholds.

Lines 356-360 in Results section: “Sensitivity analyses restricted to exposures with at least three instruments (Methods), indicated that results were largely robust across approaches (Supplementary Table S14), although significant instrument heterogeneity was observed for APOA5 in relation to MetS. Leave-one-out analysis, however, suggested that the overall causal effect was not driven by any single variant (data not shown).”

Lines 827-832 in Method section: “To ensure the robustness of our findings, we performed sensitivity analyses of the MR estimates using MR-Egger⁹ and weighted median estimation¹⁰. For proteins with more than three instruments available, instrument heterogeneity was assessed using Cochran’s Q ¹¹. Horizontal pleiotropy was assessed using MR-Egger⁹. The generalized weighted least squares (GWLS) was performed as previously described¹², while other MR analyses were performed using the TwoSampleMR¹³ R package.”

3) Colocalization methodology. The authors may wish to consider the application of more advanced colocalization approaches capable of modelling multiple causal variants, such as SuSiE or PwCoCo (e.g. PMID: 35452592). These methods are better suited to complex loci and may yield more reliable insights regarding shared genetic aetiology between proteins and traits.

Response: In response to the Reviewer’s suggestion, we applied the SuSiE approach. We note, however, that the MR results were largely consistent with those obtained using our earlier colocalization method (see also Reviewer 2’s related comment below). Compared with the original colocalization analysis, many loci did not pass the SuSiE fine-mapping step, nearly two-thirds in total, and thus could not be tested for colocalization. For this reason, we present both the original

analysis and the SuSiE-based results, now shown in the updated Table S13 (formerly Table S12). The description in the Methods section has been revised.

Lines 801-819 in Method section:

“Colocalization and Mendelian randomization analysis

We used colocalization analysis to identify cis-acting pQTLs that shared a common signal with six different phenotypic traits using the R package coloc (version 5.2.3)¹⁴ and publicly available GWAS summary statistics from individuals with European ancestry (Supplementary Table S10). We first identified all proteins that had a cis-acting pQTL (FDR < 5%) that shared at least 1 common cis-pSNP with the GWAS trait ($P < 5 \times 10^{-8}$) and extracted the summary statistics for all SNPs within ± 150 Kb of the transcription start site for the cognate encoding gene. We then estimated the probability of a shared causal variant (PP.H4) using approximate Bayes Factor colocalization⁵ via the coloc.abf (coloc) function using a threshold of $PP > 0.9$ and > 0.5 . Additionally, we tested colocalization between overlapping signals of cardiometabolic GWAS traits and serum protein cis-acting pQTL using the Bayes Factor colocalization⁵ with Sum of Single Effects (SuSiE) framework for fine mapping of genomic loci⁶. The SNPs within the cis window were fine mapped for both the GWAS and pQTL signal using the susie_rss() function from the SusieR R package (version 0.12.35), while limiting the number of possible independent signals ($L = 5$) and the maximum number of iterations ($n = 5,000$). Only loci where convergence was observed for both signals were taken forward for colocalisation. Colocalization was performed using the coloc.susie() function from the Coloc R package (version 5.2.3), which performs Bayes Factor colocalization. We selected for colocalized signals (H4) using the posterior probability thresholds > 0.9 and > 0.5 .”

4) Inclusion of positive controls. It may be instructive to include a small number of well-established biological pathways or protein complexes—such as those involved in transcriptional regulation, the cell cycle, or intracellular transport—as positive controls. Demonstrating that the network approach successfully identifies such relationships would lend confidence to the method's validity.

Response: We appreciate the reviewer's suggestion to include well-established biological pathways or protein complexes as positive controls to validate our network approach. In the current version of the manuscript, we have indeed identified edges in the causal serum protein network that overlap with direct physical protein–protein interactions (PPIs) across multiple confidence thresholds using the HIPPIE database (z-score = 24.3, $P < 0.001$). These overlaps serve as a form of external validation, demonstrating that our network captures biologically plausible interactions.

5) Ancestry and generalizability As the study sample is limited to individuals of European descent, the authors should more clearly acknowledge this as a limitation. While expansion to more diverse populations may not yet be feasible, explicit recognition of this issue will help orient future work towards greater inclusivity and global relevance.

Response: We thank the reviewer for highlighting this point. In response, we have included a new sentence in the Discussion section, where we had previously addressed other limitations of our study.

Lines 497-500 in the Discussion section: “*First, all AGES participants are of European descent (i.e., white/Caucasian), which may limit the generalizability of our findings, as protein predictors and clinical indicators of heart disease can differ across populations with diverse genetic and environmental backgrounds.*”

Minor comment

Line 540: It would be helpful to define the symbol “E” for the benefit of readers less familiar with network analysis terminology.

Response: We thank the reviewer and have revised the text in the Methods section (Line 648) to clarify the meaning of *E*. Specifically; *E* represents the *cis*-acting pQTL instrument in the causal pathway $E \rightarrow A \rightarrow B$.

In sum, this is a high-quality manuscript addressing an important topic, with clear translational relevance. The above suggestions are intended to strengthen the rigour and transparency of the work. I look forward to seeing the revised version.

Reviewer #2:

The manuscript presents a sophisticated causal network inference framework leveraging 7,523 serum proteins measured in 5,376 elderly individuals from the AGES-Reykjavik Study. The rationale for a system-level approach and the use of pQTLs for causal protein network inference is well-articulated, and the methodology is technically sound but highly complex. Below are specific concerns and suggestions for improvement.

We thank the reviewer for his/her positive remarks.

1. Abstract

The statement "we used cis-acting serum protein quantitative trait loci (pQTLs) as instrumental variables, while accounting for potential unobserved confounding factors" requires clarification. If this refers to Mendelian randomization (MR), the phrasing is misleading—MR mitigates unobserved confounding via genetic IVs but does not explicitly "account" for it. Please revise to reflect MR's assumptions accurately.

Response: We thank the Reviewer for this helpful comment. We agree that the original phrasing could be misleading in the context of MR, as our statement referred specifically to the methodology used for network reconstruction within the *Findr* framework. To avoid confusion, we have revised the Abstract accordingly. The sentence now reads:

"To reconstruct a global causal network of circulating proteins, we used cis-acting serum protein quantitative trait loci (pQTLs) as instrumental variables within a causal inference framework, which includes tests designed to mitigate the influence of hidden confounders."

2. Results (Line 129–130)

"The follow-up period for newly diagnosed MI patients extended up to 12 years from baseline, with recurrent cases excluded from the incident group analysis. Person-years of follow-up were calculated from the first AGES visit until the date of diagnosis, death, or the end of the follow-up period..." Incident vs. Recurrent Cases: If follow-up was censored at the first MI diagnosis (i.e., incident cases only), recurrent MI cases should not exist in this cohort by definition.

Response: We thank the Reviewer for his/her observation and would like to clarify that follow-up was indeed censored at the time of the first MI diagnosis, ensuring that only first (incident) MI events were included. By design, recurrent MI cases were not part of the incident cohort, and we have therefore removed any reference to recurrent MI in the corresponding sentence (Lines 130-133).

3. Discussion

The clinical implications of prioritized proteins (Table 2) are underdeveloped. Given the focus on ASCVD early detection and prevention, readers would benefit from:

- A brief summary of how many proteins have licensed/investigational drugs targeting them.
- Potential therapeutic pathways suggested by the causal network.

Response: We fully agree with the Reviewer and, in response, have generated a new Supplementary Table S16 reporting whether each top-ranked network regulator is druggable or targeted by existing compounds using public databases, and have added corresponding description to the Supplementary Text (new section called *Therapeutic pathways and clinical implications*) and Results and Discussion sections to highlight potential therapeutic pathways and clinical implications. In response to related comments from Reviewer 3, we used information from Supplementary Table S16 and another newly added Supplementary Table S15 (see related comment by Reviewer 3 below) to generate a new Table 3 in the main text. This table expands on additional links between the regulators and ACVD and integrates directional consistency across causal, observational, and therapeutic evidence for each regulator.

Lines 404-426 in Results: *“Therapeutic pathways and clinical implications: Over half of the top-ranked network regulators are potentially druggable with small molecules or biologics, although only a few have compounds currently available for either related or unrelated disease indications (Supplementary Table S16, Supplementary Text). Specifically, 13 of the 25 regulators are druggable, with 6 already targeted by licensed or investigational drugs. Building on this, the CPN analysis has identified both interconnected networks and therapeutic pathways with potential for early ACVD detection and prevention: 1) Complement-mediated inflammation: The network regulators C2 and CFB are central to the alternative complement pathway, driving vascular inflammation and atherosclerosis¹⁵. CFB is targeted by the antisense oligonucleotide Sefaxersen, in development for IgA nephropathy (Supplementary Table S16). Targeting this pathway could prevent early inflammatory progression of atherosclerotic lesions. 2) Cellular stress response and cyto-protection: KEAPI regulates the Nrf2 oxidative stress pathway¹⁶, while HSPA1A and HSPA1B control protein folding under stress¹⁷. All three have licensed drugs targeting them (Supplementary Table S16). Modulating this pathway may protect against oxidative damage and endothelial dysfunction. 3) Hemostasis: The key target KLKB1, a regulator of plasma kallikrein in the intrinsic coagulation pathway¹⁸, is targeted by the approved drug Berotralstat against angioedema (Supplementary Table S16). Our integrative analysis, however, implies that activating KLKB1 may offer a strategy to prevent thrombotic complications in ACVD (Table 3). 4) Lipid metabolism: The network regulators APOA5 and AFM control triglyceride-rich lipoprotein metabolism and oxidative stress¹⁹⁻²¹, are bio-druggable (Supplementary Table S16), and could be targeted to prevent dyslipidemia-driven initiation of atherosclerosis. This network framework highlights causal pathways for ACVD prevention, with concordant data strengthening target confidence and inconsistencies pointing to opportunities for novel therapeutic discovery.”*

Lines 519-525 in Discussion: *“Indeed, among 25 network regulators, 13 are druggable, with 6 already targeted by licensed or investigational drugs. Accordingly, the CPN reveals interconnected therapeutic pathways, including complement-mediated inflammation, cellular stress response, hemostatic balance, and lipid metabolism, supporting both single- and multi-pathway targeting. This framework provides a roadmap for next-generation ASCVD prevention, where concordant evidence reinforces target confidence and discrepancies highlight opportunities for novel therapeutic discovery.”*

4. Methods

Clarify definition of NAFLD/MASLD criteria.

Response: We have revised the Methods section to clarify the diagnostic criteria for NAFLD/MASLD.

Lines 583-585 in Method section: *“Hepatic steatosis was assessed by computed tomography (CT), serving as a non-invasive proxy for non-alcoholic fatty liver disease (NAFLD), as previously described²²”*

Genomic Window Inconsistencies: Varying window sizes are used across analyses (300 kb for cis-acting pQTL detection, 150 kb for variance modeling/colocalization, 500 kb for MR IVs). Justify these choices.

Response: We thank the reviewer for highlighting this point. To clarify, the 300 kb window used for *cis-acting pQTL* detection is equivalent to the ± 150 kb window applied in the variance modeling and colocalization analyses, i.e., both define the same 300 kb genomic span centered on the gene. For the Mendelian randomization analyses, a broader window was used to define *cis*-instruments, based on individual disease studies conducted independently of the network reconstruction and the *cis*-window applied therein. For consistency we have repeated the MR analysis using a 300 kb (± 150 kb) window and updated the text accordingly (see track changes version of the manuscript). Please see our response to the same reviewer's comment on LD clumping thresholds below. Updated results are presented in Table S13 (previously Table S12).

Colocalization Assumptions: The analysis assumes a single causal variant. For robustness, consider methods accommodating multiple causal variants (e.g., PWCoCo, SharePro) for prioritized proteins.

Response: Please see our response to a similar comment from Reviewer 1 (comment 3), who suggested using SuSiE. Following the recommendations of both reviewers, we revisited the colocalization analysis with an additional approach. The updated results, alongside the previous analyses, are presented in Table S13 (formerly Table S12), and the Methods section has been revised accordingly.

Lines 801-819 in Method section:

“Colocalization and Mendelian randomization analysis

We used colocalization analysis to identify cis-acting pQTLs that shared a common signal with six different phenotypic traits using the R package coloc (version 5.2.3)¹⁴ and publicly available GWAS summary statistics from individuals with European ancestry (Supplementary Table S10). We first identified all proteins that had a cis-acting pQTL (FDR < 5%) that shared at least 1 common cis-pSNP with the GWAS trait ($P < 5 \times 10^{-8}$) and extracted the summary statistics for all SNPs within ± 150 Kb of the transcription start site for the cognate encoding gene. We then estimated the probability of a shared causal variant (PP.H4) using approximate Bayes Factor colocalization⁵ via the coloc.abf (coloc) function using a threshold of $PP > 0.9$ and > 0.5 . Additionally, we tested colocalization between overlapping signals of cardiometabolic GWAS traits and serum protein cis-acting pQTL using the Bayes Factor colocalization⁵ with Sum of Single Effects (SuSiE) framework for fine mapping of genomic loci⁶. The SNPs within the cis window were fine mapped for both the GWAS and pQTL signal using the susie_rss() function from the SusieR R package (version 0.12.35), while limiting the number of possible independent signals

(L = 5) and the maximum number of iterations (n = 5,000). Only loci where convergence was observed for both signals were taken forward for colocalisation. Colocalization was performed using the coloc.susie() function from the Coloc R package (version 5.2.3), which performs Bayes Factor colocalization. We selected for colocalized signals (H4) using the posterior probability thresholds > 0.9 and >0.5.”

The LD clumping threshold ($r^2 \geq 0.2$) is lenient and may retain non-independent cis-acting pQTLs. Address whether stricter thresholds (e.g., $r^2 < 0.1$) is not feasible here.

Response: In addition to narrowing the genomic window in the MR to ensure consistency across analyses (see response above), we applied a more stringent LD clumping thresholds of $r^2 < 0.1$ in addition to the original threshold $r^2 < 0.2$. We note however that the GWLS estimator used for the MR analysis accounts for correlation between instruments²³. Overall, the main findings remain broadly consistent, with some associations reaching higher levels of significance. The updated results are now provided in Table S13 (previously Table S12).

Reviewer #3:

Summary

This manuscript by Emilsson and colleagues presents a comprehensive systems biology approach to understand the molecular outcomes underlying myocardial infarction (MI) and associated cardiometabolic conditions. By analyzing serum proteomic data through Somascan with samples from the AGES-Reykjavik study and employing a causal inference framework using cis-acting pQTLs, the authors reconstruct 185 high-confidence causal protein subnetworks (CPNs). These subnetworks were then associated with incident MI, metabolic syndrome, heart failure, and related traits. The authors attempted to validate their network using internal metrics and reference protein-protein interaction databases and lastly performed colocalization and Mendelian Randomization (MR) analyses in order to strengthen their claims.

I think that this study represents an important advancement in causal proteomics and has the potential to uncover novel biomarkers and therapeutic targets for cardiovascular disease. The greater than 5,000 samples afforded by the AGES cohort is compelling. However, some conceptual and methodological weaknesses should be addressed to enhance the study's impact and interpretability. Importantly, the analytical framework is strong, but the study would benefit from independent validation, greater biological resolution, and experimental follow-up.

Major Strengths

The AGES-Reykjavik cohort provides deep phenotyping and proteomic data, with sufficient statistical power to explore complex biological relationships because of this large, richly annotated patient cohort: This is considered a high profile strength of the project

The application of Mendelian Randomization and the Findr pipeline to reconstruct directed protein networks from high-dimensional serum data is well-justified and methodologically sound. And represents an innovative use of causal inference.

The authors conduct extensive robustness tests, including AUC/precision-recall analysis, transitive reduction, and overlap with known PPIs, lending feasibility and credibility to their findings. Associations with future MI and HF events, alongside known clinical traits, support the clinical applicability of the identified network regulators (e.g., ITIH3, APOA5) showing translational potential.

We appreciate the reviewer's positive feedback.

Major Weaknesses and Areas for Improvement

Lack of External Validation. The causal protein networks are derived from a single cohort of elderly Icelandic participants. No independent replication is attempted in other populations or platforms. It is entirely possible to validate key subnetworks or associations (e.g., ITIH3, DDX39B, APOA5) in an external proteomic cohort even if limited in number, or more relevant to a multi-ethnic cohort. It is entirely possible as well, that there are public available datasets, such as the UK biobank which could serve as a validation informatic approach.

Response: While we previously acknowledged the current scarcity of external datasets suitable for our network -structure replication, we fully agree with the Reviewer that independent validation would strengthen the conclusion of our study. We have therefore assessed the causal protein network architecture in the UK Biobank Olink dataset, comprising 52,543 samples and 2,186 proteins, despite previously noted modest cross platform concordance with our -SomaLogic aptamer-based protein measurements (including only moderate Spearman correlations and discrepancies in *cis-acting pQTL* signals for many analytes)¹. Accordingly, while performing this analysis, we have made a point of emphasizing platform heterogeneity as an important caveat for interpreting cross-cohort comparability. Based on this analysis, we have added new Supplementary Table S10 and Supplementary Fig. S11 (displayed at the end of the document), along with corresponding text in the Results, Discussion, and Methods sections as detailed below:

Lines 297-337 in the Result section:

“Replication of causal protein network architecture in an independent cohort

“We performed a validation analysis of the CPN structure using the UK Biobank (UKBB) study² as an independent dataset. There are 2,186 assays from the UKBB Olink Explore and the AGES 7K SOMAScan profiling platforms that target the same proteins according to their annotations (Methods). Therefore, we reconstructed CPNs in both AGES and UKBB using only this subset of proteins from each platform to ensure that these networks were comparable. As correlations in protein measurements between these two different proteomic platforms have been shown to vary³, we conducted instrument selection separately in the UKBB and AGES studies (Methods), removing duplicate aptamers targeting the same protein before analysis.

*We first reconstructed a subsetted CPN in AGES but restricted it to the 2,186 proteins also measured in UKBB. Of these protein assays, 1,835 had a valid *cis-acting pQTL* instrument to be used as a potential network regulator. At a 1% FDR threshold we identified 454 proteins with at least one target and 43 with at least 10 targets in the AGES study. Among the 454 proteins with at least one target in the subsetted AGES CPN (FDR = 1%), 416 were also present in the full AGES CPN constructed from all measured proteins. Furthermore, 96.3% of the edges identified in the subsetted AGES CPN overlapped with those in the full CPN. We then reconstructed a comparative CPN using data from 52,543 UKBB participants with Olink assays targeting the same 2,186 proteins. We observed a significant overlap between network regulators in the two cohorts, including the 454 proteins with at least one target in AGES, of which 393 were present with at least one target in UK Biobank (Fisher’s Exact Test; OR = 1.9, $P = 4.1 \times 10^{-6}$). Further, we identified 629 edges shared between the AGES subsetted and UK Biobank CPNs (FDR = 1%). To formalize this comparison, we applied Fisher’s Exact Test, which demonstrated that the overlap was highly significant (OR = 2.9, $P = 1.6 \times 10^{-99}$), using all possible edges as the background while excluding self-edges. Notably, many more targets were identified for a given CPN in UKBB than in AGES (Supplementary Table S10), likely to reflect the nearly tenfold larger sample size in UKBB.*

We next sought to determine the extent to which protein targets of subnetworks reconstructed in AGES and those from UKBB overlapped. Of the 393 shared network regulators from the comparative subnetworks, we found 36 that shared at least 2 targets in the AGES and UKBB comparative CPNs, for which we assessed overlap significance with a hypergeometric test, using the full set of 2,186 proteins as background. Overall, 23 subnetworks had a significant overlap in targets across both platforms ($q < 0.05$), including three of the top-ranked subnetworks previously

linked to ACVD in the full AGES CPN, namely, KLKBI, CFB and CSF3 (Supplementary Table S10, Supplementary Fig. S11).

A recent study compared correlations between Olink and SOMAlogic platforms in the Icelandic population³. From this study, we obtained correlations for 1,677 of the 2,186 proteins measured (mean Pearson coefficient = 0.39). For 19 of the 23 significantly overlapping subnetworks, correlations were available, with a higher mean of 0.48 (Supplementary Table S10). In some cases, network regulators of significantly overlapping CPNs were not correlated across platforms (Supplementary Table S10), suggesting that even when different affinity-based technologies bind distinct epitopes, the network regulator remains robust and continues to capture the same underlying biology reflected in their downstream targets.”

Lines 478-493 in the Discussion section: *“Validating our CPNs is inherently challenging because complementary external data are scarce, unlike tissue-specific gene regulatory networks that can be supported by gene expression QTLs and external transcription factor-target datasets⁴. To address this challenge, we drew on existing resources such as the UKBB^{5,6}, although its protein coverage is considerably smaller than in our study, and inter-platform correlations are modest (mean $r = 0.39$), with additional differences arising from the epitope targeting inherent to affinity-based technologies. Nevertheless, the identification of 23 significantly overlapping subnetworks (out of 36 comparable networks) across AGES and UKBB highlights that the CPN architecture is robust. We further observed strong cross-cohort replication of global regulators and their connections, with regulators and edges significantly overlapping beyond chance expectations. The UKBB’s tenfold larger sample size facilitated the detection of more targets per regulator, but its reduced proteomic coverage limited a more comprehensive replication of networks, emphasizing the dual importance of sample size and proteomic breadth for network reconstruction and comparison. Importantly, several of the top-ranked subnetworks, including KLKBI, CFB, and CSF3, were consistently replicated across platforms and populations, underscoring their biological relevance beyond platform-specific effects.”*

Lines 774-799 in Method section:

“Validating the network structure across proteomic platforms and cohorts

The UKBB is a prospective population study of 502,128 individuals from the UK who have been extensively characterized for genomic and phenotypic traits². Genotype data is available for 486,593 participants, obtained using either the Applied Biosystems UK BiLEVE Axiom Array by Affymetrix or the Applied Biosystems UKBB Axiom Array. Imputation was carried out using the HRC reference panel, yielding allelic dosages for ~96 million variants, with markers annotated using the GRCh37 assembly of the human genome⁵.

There are 53,013 (53.9% female, mean age = 56.8) UKBB participants who have undergone plasma protein profiling for 2,923 proteins using the antibody affinity based Olink Explore 3072 protein extension assay platform⁶. We processed the Olink assay data using the approach outlined by Sun et al.⁶, from the initial study reporting plasma measurements from UKBB, whereby assays with more than 20% missingness are removed and the remaining missing values are mean imputed across samples. The overlap between processed samples and those with corresponding genotype data yielded 52,543 samples to be taken forward for network reconstruction. The remaining protein assays were then filtered for those that had a corresponding assay in the AGES SOMAScan

7K panel, yielding 2,186 comparative protein assays. These assay measurements were then corrected for age and sex by fitting to a linear model as previously described for the AGES proteomic data.

Due to differences between Somalogic and Olink platforms, instrument selection was carried out independently for the UKBB proteomic data. For each protein, the lead cis-acting pSNP was chosen based on the P-value, using pQTL summary statistics from Sun et al.⁶. Reconstruction of causal protein networks was carried out with Findr and edges selected based on FDR thresholds, as has been previously described for the AGES discovery cohort. We tested the significance of overlapping targets between comparative subnetworks by hypergeometric testing using the hypergeom() function from Scipy Stats⁷ in Python, while using the intersection of measured proteins between platforms as a background. We then corrected for multiple testing using Storey-Tibshirani procedure for q-value estimation⁸.

Informatic analysis/Indirect Edge Ambiguity and Confounding interpretation. While transitive reduction is applied to remove some indirect edges, a large number of network edges may still reflect indirect regulation or residual confounding. A recommendation might be to incorporate formal mediation models or latent confounder correction to further refine causal edge specificity.

Response: Mediation analysis was already incorporated within the Findr pipeline, and we now present the results at a stringent 1% FDR threshold in a new Supplementary Table S10, which also includes the replication results in the UKBB dataset. However, we note that a recognized limitation of mediation analysis is its susceptibility to confounding, which can lead to a high rate of false negatives, a challenge that is particularly pronounced in biological networks where coregulation by shared upstream factors frequently occurs.

Functional Validation Missing.

The study is computational in nature; no experimental follow-up or perturbation assays are included to confirm network-derived predictions. Here, a suggestion would be to target a subset of high-priority regulators (e.g., ITIH3, HSPA1A, LRP4) for validation using in vitro or in vivo models, or at minimum, review existing functional studies in greater detail.

Response: Please see our responses to Reviewer 1 (comment 4), Reviewer 2 (comment 3), and the current Reviewer's comment for related discussion below (related to effect size/direction). In brief, we have expanded on the connections among the top-ranked network regulators by integrating both prior and new knowledge, along with additional interpretation, including effect direction, links to survival outcomes, established biology, and potential druggability (including licensed or investigational drugs). As detailed elsewhere, this has resulted in expanded text in the main manuscript and Supplementary Text, two new Supplementary tables, and a new Table 3 in the main text. Please see our detailed response below. However, given the scope, objectives, and design of the present study, we believe that experimental validation *in vitro* or *in vivo* is more appropriately addressed in a separate follow-up investigation and is beyond the scope of this work.

There is some potential ambiguity in protein-aptamer mapping, particular due to the platform that the authors used. Somascan has some inherent technical limitations. In this study, several aptamers map to the same protein family, sometimes targeting different isoforms or protein domains. Such dual identification could result in artificial interpretation or duplication of network regulators. It is

recommended that authors clarify how redundancy in aptamer targeting was handled and consider collapsing overlapping aptamers to reduce spurious edges.

Response: This was indeed addressed. For example, on former page 7 (current Lines 156-160) in the Results section: “For A-proteins with multiple aptamers, we selected those with the largest number of targets, and further refinement of linkage disequilibrium (LD) among *cis*-acting instruments resulted in the final CPN comprising 185 A-proteins (referred to here as network regulators), their corresponding subnetworks, and a total of 31,358 edges (Supplementary Table S1, Fig. 2A)”. In current Lines 668-670 of the Methods section, we clarified this point as well: “In instances where there were multiple aptamers targeting the same protein, we selected the A proteins with the largest number of targets as the representative aptamer for this protein.”

Phenotypic Associations and Effect Size Interpretation. While statistically significant, the magnitude of associations between network eigenproteins and traits is not always clinically contextualized. The authors are suggested to maybe report hazard ratios or beta coefficients with interpretation to support translational relevance.

Response: This is a great point which we have addressed in the context of related comments and revisions described above. Specifically, please refer to our earlier response to your comment, as well as Reviewer 2’s comment 3. We agree that interpreting both effect size and direction is critical for translational relevance and acknowledge that this was not fully addressed in the previous version of the study. It is important to note that eigenproteins represent aggregate signals derived from multiple proteins within a module. Consequently, their associations with traits often exhibit weak or modest effect sizes, reflecting average variation across the module rather than strong effects from individual proteins, such as for specific network regulators. Therefore, effect sizes cannot be directly compared between network regulators and their corresponding eigenproteins. Nonetheless, we consider the effect direction of network regulators, together with their consistency with other evidence including causal inference and associations with clinical outcomes and survival to be the most critical and note that this was overlooked in the previous version of our paper. We have now addressed this in the revised manuscript through the new Supplementary Table S15 and new Table 3 (supported by data from Supplementary Tables S15 and S16) in main text, as well as updates to the Results section and Supplementary text. For example, the Results section titled “***Additional links between top-ranked network regulators and ACVD***” has been rewritten to reflect these updates, as shown below. The new Table 3 is shown at the end of this document.

Lines 339-403 in Result section:

“Additional links between top-ranked network regulators and ACVD

We investigated several complementary lines of evidence to support the role of the top-ranked network regulators in the pathophysiology of MI and related cardiometabolic traits. Network regulators and targets from the top-ranked subnetworks showed enrichment in multiple pathways previously linked to ACVD, including a shared enrichment in the cellular response to heat shock (Supplementary Table S11, Supplementary Figs. S12–S13, Supplementary Text). STRING-based analysis revealed significantly (P -value = 0.047) enriched functional and physical interactions among the top network regulators (Supplementary Fig. S14), including previously found (Fig. 6C) and novel connections (e.g., ITIH3 with KLKB1, APOA5, and AFM; Supplementary Fig. S14).

Causal inference analysis of top-ranked regulators: Two-sample MR and colocalization analyses highlighted several top-ranked regulators including APOA5, DDX39B, HSPA1A, C11orf49 and LRP4, with significant support for causal associations with ACVD-related traits (FDR < 0.05; Supplementary Tables S12–S13). Here, APOA5 demonstrated a strong protective effect against MI and MetS, consistent with findings that APOA5 knockout mice display a fourfold elevation in plasma triglyceride levels¹⁹, a significant risk factor for ACVD²⁴. Further, genetically determined levels of HSPA1A showed an association with MetS, indicating a causal role, whereas DDX39B was linked to both MetS and T2D. Notably, the HSPA1A gene has also been causally implicated in T2D and its microvascular complications in prior MR and colocalization analyses²⁵. Sensitivity analyses restricted to exposures with at least three instruments (Methods), indicated that results were largely robust across approaches (Supplementary Table S14), although significant instrument heterogeneity was observed for APOA5 in relation to MetS. Leave-one-out analysis, however, suggested that the overall causal effect was not driven by any single variant (data not shown).

Additional context from current and prior evidence: Beyond the analyses of the present study, several top regulators, including ITIH3, HSPA1B, KEAP1, HSPA1A, C2, CSF3, FABP3, KLKB1, AFM, APOA5, PTPN11, CFB, and COL28A1, have previously been linked to ACVD or related cardiometabolic traits (Supplementary Text). For example, the top-ranked regulator ITIH3 has a gain-of-function genetic association with increased risk of MI and is expressed in vascular smooth muscle cells and macrophages within atherosclerotic plaques²⁶. Notably, ITIH3 is positioned at the base of the data-driven CPN (Fig. 3), indicating a potential role in numerous regulatory pathways. Another example is KLKB1, which we recently demonstrated through MR analysis to be causally protective against calcific aortic valve disease²⁷, a leading cause of heart failure via aortic stenosis²⁸. To further characterize the top-ranked protein network regulators, we assessed their effect directions on ACVD-related traits and survival outcomes (Supplementary Table S15) and evaluated if they are druggable or targeted by available compounds using public databases (Supplementary Table S16). Table 3 summarizes the accumulated findings, offering an integrated overview of directional consistency across causal, observational, and therapeutic evidence for each regulator.

Many of the top-ranked regulators showed consistent directionality across multiple lines of evidence and exhibit positive and often causal effects on ACVD traits (Table 3), including ITIH3, HSPA1A, DDX39B, FABP3, PTPN11, CFB, and COL28A1, whereas APOA5 and KLKB1 show protective effects (Table 3, Supplementary Text). In contrast, network regulators such as KEAP1, C2, and CSF3 show inconsistent effect directions between the present findings and previously reported associations (Table 3). KEAP1 regulates the NRF2 pathway, essential for oxidative balance and cardiovascular protection (Supplementary Text). Cardiomyocyte-specific Keap1 knockout upregulates NRF2 and prevents induced cardiac dysfunction¹⁶, contrasting with the protective association of circulating KEAP1 in our study (Table 3). This discrepancy may reflect differences between tissue-specific and circulating biology, where serum KEAP1 could indicate systemic processes independent of NRF2 inhibition or a feedback marker of activated antioxidant responses rather than a direct protective factor. A human study found C2 deficiency to be associated with increased atherosclerosis risk²⁹, suggesting a protective role for C2 (see Supplementary Text). In contrast, our finding that higher circulating C2 predicts increased incident MI risk (Table 3), may reflect differences between lifelong deficiency and elevated plasma levels, with the latter potentially indicating complement activation or inflammation. Alternatively, elevated C2 in the present study could represent a compensatory upregulation in response to

subclinical vascular injury. Further, cardiac myocytes express CSF3 under both normal and stress conditions, particularly following ischemic injury, where it supports cardiomyocyte survival after a heart attack³⁰. Notably, the positive association between circulating CSF3 levels, incident MI, and reduced survival (Table 3), may reflect a compensatory or adaptive response to early cardiac injury or stress during subclinical disease (Supplementary Text). Finally, given the conflicting findings for C11orf49 and LRP in our study and the inconsistent evidence for AFM in previous research (Table 3), the direction of their effects on ACVD etiology remains inconclusive. In summary, 68% (17 of 25) of the top-ranked network regulators, this study, along with others, reveals additional specific connections to ACVD-related traits. These findings strengthen evidence for the causal role of many of the network regulators and their targets in ACVD development and may guide therapeutic strategies to modulate them.”

Lines 514-519 in Discussion section: *“Our integrative analyses support a key mechanistic role for many top-ranked network regulators in ACVD, with mostly consistent associations observed across complementary evidence. Some regulators, including KEAP1, C2, and CSF3, showed directionally inconsistent associations, likely reflecting differences between circulating and tissue-specific biology or compensatory responses. Overall, these findings reinforce the causal relevance of key network regulators and their targets, highlighting potential avenues for therapeutic intervention.”*

Minor Point

The study is missing “Negative Controls”. For instance, it would strengthen the interpretation to report null associations (e.g., subnetworks NOT linked to MI or metabolic traits) to contextualize specificity, this is important context to the manuscript.

Response: This is an important point, particularly given that the network reconstruction process is disease-agnostic. Indeed, many of the 185 subnetworks including 62 network regulators and 121 eigenproteins, exhibited only one or no significant associations with ACVD-related traits (Supplementary Tables S5–S6). For instance, VTN, the regulator of the largest network comprising 3,632 targets, showed no significant association with any trait (Supplementary Table S5). Accordingly, we have added the following sentence to the Results section:

Lines 252-255 in Results section: *“We note that a substantial number of the 185 subnetworks, including 62 network regulators and 121 eigenproteins, showed only one or no significant associations with ACVD-related traits (Supplementary Tables S5–S6).”*

References

1. Rooney, M.R., *et al.* Plasma proteomic comparisons change as coverage expands for SomaLogic and Olink. *medRxiv* (2024).
2. Sudlow, C., *et al.* UK biobank: an open access resource for identifying the causes of a wide range of complex diseases of middle and old age. *PLoS Med* **12**, e1001779 (2015).
3. Eldjarn, G.H., *et al.* Author Correction: Large-scale plasma proteomics comparisons through genetics and disease associations. *Nature* **630**, E3 (2024).
4. Gerstein, M.B., *et al.* Architecture of the human regulatory network derived from ENCODE data. *Nature* **489**, 91-100 (2012).
5. Bycroft, C., *et al.* The UK Biobank resource with deep phenotyping and genomic data. *Nature* **562**, 203-209 (2018).
6. Sun, B.B., *et al.* Plasma proteomic associations with genetics and health in the UK Biobank. *Nature* **622**, 329-338 (2023).
7. Virtanen, P., *et al.* SciPy 1.0: fundamental algorithms for scientific computing in Python. *Nat Methods* **17**, 261-272 (2020).
8. Storey, J.D. & Tibshirani, R. Statistical significance for genomewide studies. *Proc Natl Acad Sci U S A* **100**, 9440-9445 (2003).
9. Burgess, S. & Thompson, S.G. Interpreting findings from Mendelian randomization using the MR-Egger method. *Eur J Epidemiol* **32**, 377-389 (2017).
10. Bowden, J., Davey Smith, G., Haycock, P.C. & Burgess, S. Consistent Estimation in Mendelian Randomization with Some Invalid Instruments Using a Weighted Median Estimator. *Genet Epidemiol* **40**, 304-314 (2016).
11. Cochran, W.G. The Combination of Estimates from Different Experiments. *Biometrics* **10**, 101-129 (1954).
12. Jonmundsson, T., *et al.* A proteomic analysis of atrial fibrillation in a prospective longitudinal cohort (AGES-Reykjavik study). *Europace : European pacing, arrhythmias, and cardiac electrophysiology : journal of the working groups on cardiac pacing, arrhythmias, and cardiac cellular electrophysiology of the European Society of Cardiology* **25**(2023).
13. Hemani, G., *et al.* The MR-Base platform supports systematic causal inference across the human phenome. *Elife* **7**(2018).
14. Giambartolomei, C., *et al.* Bayesian test for colocalisation between pairs of genetic association studies using summary statistics. *PLoS Genet* **10**, e1004383 (2014).
15. Liao, C.C., Xu, J.W., Huang, W.C., Chang, H.C. & Tung, Y.T. Plasma Proteomic Changes of Atherosclerosis after Exercise in ApoE Knockout Mice. *Biology* **11**(2022).
16. Zoccarato, A., *et al.* NRF2 activation in the heart induces glucose metabolic reprogramming and reduces cardiac dysfunction via upregulation of the pentose phosphate pathway. *Cardiovascular research* **121**, 339-352 (2025).
17. Hess, K., *et al.* Concurrent action of purifying selection and gene conversion results in extreme conservation of the major stress-inducible Hsp70 genes in mammals. *Sci Rep* **8**, 5082 (2018).
18. Wan, J., *et al.* Kallikrein augments the anticoagulant function of the protein C system in thrombin generation. *Journal of thrombosis and haemostasis : JTH* **20**, 48-57 (2022).
19. Pennacchio, L.A., *et al.* An apolipoprotein influencing triglycerides in humans and mice revealed by comparative sequencing. *Science* **294**, 169-173 (2001).
20. Kronenberg, F., *et al.* Plasma concentrations of afamin are associated with the prevalence and development of metabolic syndrome. *Circ Cardiovasc Genet* **7**, 822-829 (2014).
21. Nowicki, G.J., Ślusarska, B., Polak, M., Naylor, K. & Kocki, T. Relationship between Serum Kallistatin and Afamin and Anthropometric Factors Associated with Obesity and of Being Overweight in Patients after Myocardial Infarction and without Myocardial Infarction. *Journal of clinical medicine* **10**(2021).

22. Speliotes, E.K., *et al.* Genome-wide association analysis identifies variants associated with nonalcoholic fatty liver disease that have distinct effects on metabolic traits. *PLoS Genet* **7**, e1001324 (2011).
23. Gkatzionis, A., Burgess, S. & Newcombe, P.J. Statistical methods for cis-Mendelian randomization with two-sample summary-level data. *Genet Epidemiol* **47**, 3-25 (2023).
24. Budoff, M. Triglycerides and Triglyceride-Rich Lipoproteins in the Causal Pathway of Cardiovascular Disease. *The American journal of cardiology* **118**, 138-145 (2016).
25. Yuan, S., *et al.* Plasma proteins and onset of type 2 diabetes and diabetic complications: Proteome-wide Mendelian randomization and colocalization analyses. *Cell reports. Medicine* **4**, 101174 (2023).
26. Ebana, Y., *et al.* A functional SNP in ITIH3 is associated with susceptibility to myocardial infarction. *Journal of human genetics* **52**, 220-229 (2007).
27. Bortnick, A.E., *et al.* Plasma Proteomic Assessment of Calcific Aortic Valve Disease in Older Adults. *J Am Heart Assoc* **14**, e036336 (2025).
28. Lerman, D.A., Prasad, S. & Alotti, N. Calcific Aortic Valve Disease: Molecular Mechanisms and Therapeutic Approaches. *European cardiology* **10**, 108-112 (2015).
29. Jönsson, G., *et al.* Hereditary C2 deficiency in Sweden: frequent occurrence of invasive infection, atherosclerosis, and rheumatic disease. *Medicine (Baltimore)* **84**, 23-34 (2005).
30. Harada, M., *et al.* G-CSF prevents cardiac remodeling after myocardial infarction by activating the Jak-Stat pathway in cardiomyocytes. *Nat Med* **11**, 305-311 (2005).
31. Kurki, M.I., *et al.* FinnGen provides genetic insights from a well-phenotyped isolated population. *Nature* **613**, 508-518 (2023).
32. Zhuang, L., *et al.* Fatty acid-binding protein 3 contributes to ischemic heart injury by regulating cardiac myocyte apoptosis and MAPK pathways. *American journal of physiology. Heart and circulatory physiology* **316**, H971-h984 (2019).
33. Schachtl-Riess, J.F., *et al.* KLKB1 and CLSTN2 are associated with HDL-mediated cholesterol efflux capacity in a genome-wide association study. *Atherosclerosis* **368**, 1-11 (2023).
34. Araki, T., *et al.* Noonan syndrome cardiac defects are caused by PTPN11 acting in endocardium to enhance endocardial-mesenchymal transformation. *Proc Natl Acad Sci U S A* **106**, 4736-4741 (2009).
35. Coan, P.M., *et al.* Complement Factor B Is a Determinant of Both Metabolic and Cardiovascular Features of Metabolic Syndrome. *Hypertension (Dallas, Tex. : 1979)* **70**, 624-633 (2017).
36. Peters, A.E., *et al.* Proteomic Pathways across Ejection Fraction Spectrum in Heart Failure: an EXSCEL Substudy. *medRxiv* (2023).

Table 3. Summary of network regulators, including findings from the present study, previously known biological links, effect directions, if druggable or status as a target of an available compound

Regulator	Evidence from MR/Colocalization in AGES, and prior known biology related to ACVD*	Effect direction on ACVD and/or survival (Table S13)	Druggable / available compound (Table S15)
ITIH3	Gain-of-function variant increases risk of MI. Highly expressed in atherosclerotic lesion	Risk	Druggable
HSPA1B	Causally linked to T2D and microvascular complications	Risk	Available compound
DDX39B	Causally related to T2D and MetS (AGES)	Risk	Not druggable
HSPA1A	Causally related to MetS (AGES). Causally linked to T2D and microvascular complications	Risk	Available compound
KEAP1	Cardiomyocyte-specific knockout mice show improved cardiac function	Protective	Available compound
C2	Deficiency increases atherosclerosis	Risk	Druggable
PCDH8	None	Risk	Not druggable
GABARAP	Genetic links to hypertension and CVD	Risk	Not druggable
CSH1/CSH2	None	Protective	Druggable
RFC4	Genetic links to diabetes and venous thromboembolism	Risk	Not druggable
IZUMO1	None	Risk	Not druggable
CSF3	Promotes cardiomyocyte survival after MI	Risk	Druggable
FABP3	Contributes to ischemic heart injury	Risk	Not druggable
KLKB1	Causally protective against calcific aortic valve disease; influences cholesterol efflux	Protective	Available compound
DCTN2	None	Risk	Not druggable
AFM	Positively associated with the metabolic syndrome, but lower levels in MI patients	Protective	Druggable
AIPL1	None	Protective	Not druggable
MRRF	None	Protective	Not druggable
APOA5	Causally protective against MI and MetS (AGES). Knockout leads to elevated plasma triglycerides	Protective	Druggable
PTPN11	Gain-of-function mutations cause heart defects in Noonan syndrome	Risk	Available compound
CFB	Knock-out mice show improved metabolic and cardiovascular features	Risk	Available compound
GAL3ST1	None	Risk	Not druggable
NUDT21	None	Risk	Not druggable
COL28A1	Positively associated with HF and its subtypes	Risk	Druggable
C11orf49	Causally linked to MI (AGES)	Protective	Not druggable
LRP4	Causally linked to MI (AGES)	Inconclusive	Not druggable

*Background on previously characterized biological roles of network regulators: ITIH3²⁶; HSPA1A/B²⁵; KEAP1¹⁶; C2²⁹; GABARAP³¹; RFC4³¹; CSF3³⁰; FABP3³²; KLKB1^{27,33}; AFM^{20,21}; APOA5¹⁹; PTPN11³⁴; CFB³⁵; COL28A1³⁶.

Supplementary Fig. S11. Network visualization from the comparative CPN analysis between AGES and UK Biobank. Edges visualized were identified in both CPNs (FDR = 1%) and filtered to remove network regulators with single targets. 36 network regulators are labelled and shown in red and there are 281 unique target proteins shown in blue (some red regulators may also be targets).

Response to Review

We are pleased to submit the revised version of our manuscript, entitled “*Circulating causal protein networks linked to future risk of myocardial infarction*” (NCOMMS-25-18281-B), for consideration for publication in *Nature Communications*. We have carefully reviewed the Author Checklist and made all necessary changes, as reflected in the files and documents uploaded with the resubmission. We thank the reviewers for their positive remarks.

Reviewer comments

Reviewer #1 (Remarks to the Author):

The revised manuscript is much stronger, particularly due to the replication in UKBB and the SuSiE colocalization analysis. I have no objections to its acceptance for publication in *Nature Communications*.

Response: We appreciate the positive remarks.

Reviewer #2 (Remarks to the Author):

None.

Response: We thank the reviewer

Reviewer #3 (Remarks to the Author):

I have no additional concerns

Response: We thanks the reviewer